# Apolipoprotein E controls Dectin-1-dependent development of monocyte-derived alveolar macrophages upon pulmonary β-glucan-induced inflammatory adaptation

The lung is constantly exposed to the outside world and optimal adaptation of immune responses is crucial for efficient pathogen clearance. However, mechanisms that lead to lung-associated macrophages' functional and developmental adaptation remain elusive. To reveal such mechanisms, we developed a reductionist model of environmental intranasal β-glucan exposure, allowing for the detailed interrogation of molecular mechanisms of pulmonary macrophage adaptation. Employing single-cell transcriptomics, high-dimensional imaging and flow cytometric characterization paired with in vivo and ex vivo challenge models, we reveal that pulmonary low-grade inflammation results in the development of apolipoprotein E (ApoE)-dependent monocyte-derived alveolar macrophages (ApoE+CD11b+ AMs). ApoE+CD11b+ AMs expressed high levels of CD11b, ApoE, Gpnmb and Ccl6, were glycolytic, highly phagocytic and produced large amounts of interleukin-6 upon restimulation. Functional differences were cell intrinsic, and myeloid cell-specific ApoE ablation inhibited Ly6c+ monocyte to ApoE+CD11b+ AM differentiation dependent on macrophage colony-stimulating factor secretion, promoting ApoE+CD11b+ AM cell death and thus impeding ApoE+CD11b+ AM maintenance. In vivo, β-glucan-elicited ApoE+CD11b+ AMs limited the bacterial burden of *Legionella pneumophilia* after infection and improved the disease outcome in vivo and ex vivo in a murine lung fibrosis model. Collectively these data identify ApoE+CD11b+ AMs generated upon environmental cues, under the control of ApoE signaling, as an essential determinant for lung adaptation enhancing tissue resilience.

The lung is exposed to a variety of immunostimulatory agents shaping its immune responses[1]. How such environmental non-pathological immune activation is controlled at the cellular and molecular levels is poorly understood. β-glucans are integral components of environmental pathogenic and non-pathogenic fungi and are proposed as immune modulators[2]. Ambient concentrations of β-glucan oscillate during the year and, in combination with pathogen exposure, correlate with allergic rhinitis increases[3]. Recognition of β-glucan by

✉e-mail: andreas.schlitzer@uni-bonn.de

Dectin-1 modulates systemic immune responses through a process termed innate immune memory[4], characterized by increased cytokine responses of monocytes, upon a secondary heterologous stimulus, facilitated by a metabolic and epigenetic rewiring, allowing a more efficient first-line immune response[4–9]. Lung-specific mechanisms of β-glucan, the organ where it is most often recognized, and how it acts as a nongenetic modifier of immune responses to subsequent disease, remain unknown.

Immune cells residing in the alveolar space are the body's respiratory first line of defense, ensuring efficient immune response induction against airborne pathogens, while regulating immune activation to ensure intact lung function[10]. AMs and monocytes constitute the major mononuclear phagocytes (MPs) found during homeostasis in humans and mice within the alveolus. AMs and monocytes express high amounts of Dectin-1 and are highly plastic[11]. Dectin-1 signaling critically depends on spleen tyrosine kinase (Syk) which upon activation triggers phospholipase C gamma 2 (PLCγ2)-dependent calcium release, downstream nuclear factor of activated T cells (NFAT) and extracellular signal-regulated kinase (ERK) activation. This cascade leads to production of interleukin (IL)-2 and IL-10. Furthermore, Syk activates caspase recruitment domain-containing protein 9 (CARD9) leading to nuclear factor kappa-light-chain-enhancer of activated B cells (NF-κB) signaling and the release of tumor necrosis factor (TNF) and IL-6. Additionally, Dectin-1 ligation directly induces reactive oxygen species (ROS) production via activation of phosphoinositide 3 kinase (PI3K)[12,13]. Murine homeostatic AMs are embryonically derived with only minor contribution of adult bone marrow (BM) homeostasis[14,15]. Pulmonary viral infection or radiation was shown to induce differentiation of Ly6c+ monocytes into long-lived monocyte-derived alveolar macrophages (MoAMs)[16–18]. During viral infection, MoAM-derived IL-6 is crucial for the defense against subsequent *Streptococcus pneumoniae* infection[16]. Finally, viral-induced environmental adaptation affects resident AM-dependent CD8+ T cell rewiring, inducing efficient bacterial clearance[19].

Chronic lung diseases, like asthma or pulmonary fibrosis, hinge on environmental factors. Thus, understanding environmental immune adaptation in bronchoalveolar MPs is crucial for insights into cellular, functional and molecular consequences[20–24].

To investigate this, we developed a reductionist model of a single low-dose intranasal β-glucan exposure. Using single-cell transcriptomic, functional in vivo and ex vivo analysis of cellular development and function, this model allowed us to dissect acute and chronic molecular adaptations of macrophages upon environmental cues. We show that a single intranasal β-glucan exposure induces developmentally and functionally modified ApoE+CD11b+ MoAMs, detected up to 21 days after β-glucan exposure. ApoE+CD11b+ MoAMs are glycolytic, highly phagocytic and release, upon activation, high amounts of IL-6. Functional changes are cell intrinsic and upon subsequent infection with *Legionella pneumophila* or a challenge by bleomycin-induced fibrosis lead to improved in vivo outcomes. Molecularly, ApoE+CD11b+ MoAMs are controlled by the Dectin-1–CARD9 pathway, whereas maintenance of ApoE+CD11b+ MoAMs depends on paracrine ApoE and macrophage colony-stimulating factor (M-CSF). Taken together, we identify ApoE as a crucial checkpoint for low-grade inflammation-associated M-CSF-controlled monocyte-to-macrophage differentiation triggered by the Dectin-1–CARD9 pathway within the immune-adapted microenvironment of lung.

## Results

### Intranasal β-glucan induces ApoE+CD11b+ AMs

The lung is constantly exposed to pollutants, sterile and non-sterile pathogens, and components thereof. Our understanding of the cellular and molecular mechanisms of immune adaptation to these environmental cues is limited. To investigate this, we developed a simplified model involving a single low-dose intranasal exposure to β-glucan particles (200 μg), mimicking environmental exposure[25]. To assess the impact of this stimulation and investigate its lasting effects, we examined bronchoalveolar lavage fluid (BALF)-resident macrophages (CD45+Lin−SSC^int−hi) of C57BL/6 mice 7 days after intranasal phosphate-buffered saline (PBS) or β-glucan treatment using single-cell transcriptomics (Fig. 1a and Supplementary Fig. 1a–c). Dimensionality reduction using uniform manifold approximation and projection (UMAP) analysis and unsupervised clustering (Louvain) revealed five distinct transcriptional clusters within the BALF (Fig. 1a,b). Here, high expression of AM signature genes *Siglecf* and *Itgax* identified all investigated cells as AMs (Supplementary Fig. 1d,e)[26]. Further analysis identified clusters 0 and 1 as subsets of resident AMs expressing Ear2, Wfdc21 and Hmox1 (refs. 26,27). Cluster 2 (proliferating AM) expressed genes linked to proliferation such as *Top2a*, *Mki67* and Birc5 (ref. 28). Cluster 3 (ApoE+ AMs) expressed genes associated with a lipid-associated inflammatory monocyte-derived macrophage (MoMac) phenotype, including *Apoe*, *Cd63*, *Spp1*, *Gpnmb* and *Trem2* (refs. 16,29,30). Cluster 4 (ISG+ AMs), was characterized by expression of interferon-stimulated genes (ISGs), such as *Ifit2*, *Ifit3*, *Ifi204* and *Isg15* (Fig. 1b). To determine which cluster was associated with β-glucan-induced environmental adaptation, the relative contribution of each stimulatory condition to individual clusters was examined (Fig. 1c–e and Supplementary Fig. 1f). Here, ApoE+ AMs were only present within the BALF of β-glucan-exposed mice 7 days prior, concomitant with a reduction in proliferating AMs. Previous studies suggested that CD11b expression on stimulated AMs serves as a marker for enhanced inflammatory potential[31,32]. Therefore, we used co-detection by indexing (CODEX)-enabled high-dimensional imaging to characterize the phenotype of ApoE+ AMs at the protein level in the lung and in BALF 7 days after β-glucan stimulation (Fig. 1f and Supplementary Fig. 1g)[33]. ApoE+ AMs not only coexpressed the classical AM markers, CD11c and Siglec-F, but also expressed high amounts of CD11b, ApoE and GPNMB proteins (Fig. 1f and Supplementary Fig. 1g). Furthermore, to confirm the overlap of *Apoe* mRNA expression and CD11b protein expression, we measured *Apoe* mRNA levels using PrimeFlow. *Apoe* mRNA signals were detectable only in BALF CD11b+ AMs isolated from mice stimulated with β-glucan 7 days prior (Fig. 1g). Consequently, we refer to this AM subpopulation as

**Fig. 1 | Intranasal β-glucan exposure generates environmentally adapted ApoE+CD11b+ AMs within the bronchoalveolar space. a–e**, Single-cell RNA sequencing (scRNA-seq) of the BALF of male 8- to 12-week-old C57BL/6J (WT) mice after intranasal stimulation with 200 μg β-glucan or PBS (*n* = 10,202 cells). Seven days after exposure, cells of three mice per condition were harvested and sorted for SSC^hi, Lin− (B220, CD19, CD3ε, Nk1.1, Ter-119), DRAQ7− singlets (*n* = 3 mice, one independent experiment). **a**, UMAP analysis of both conditions combined shows five different clusters. **b**, Heat map of the top ten highly expressed genes for each of the five clusters. **c,d**, UMAP from **a** separated by PBS (**c**; *n* = 4,845 cells) or β-glucan (**d**, *n* = 5,357 cells) condition. **e**, Percentage contribution of the five annotated clusters to overall cells split by conditions. **f**, 5-μm frozen section of the left lobe of the lung of a *Ms4a3-cre^Rosa26TOMATO* mouse 7 days after β-glucan exposure stained with a 17-plex CODEX antibody panel.

Overlaid images show the markers used to identify AM populations. Single stainings of these markers are shown in grayscale. Filled arrowheads indicate ApoE+CD11b+ AMs, whereas open arrowheads indicate CD11b− AMs. Scale bar, 100 μm (large image) and 10 μm (enlargements; *n* = one representative mouse of two independent experiments). **g**, Detection of *Apoe* mRNA expression in the AM subsets of the BALF 7 days after intranasal PBS or β-glucan exposure in WT mice by PrimeFlow (*n* = 3 mice pooled per group, one of two independent experiments shown). **h,i**, Flow cytometric quantification of absolute numbers (**h**) or frequency (**i**) of ApoE+CD11b+ AM (CD45+Siglec-F+CD64+CD11c+CD11b+) in the BALF in a time course from 1 to 21 days after β-glucan stimulation of WT mice (*n* = 9–10 mice, two independent experiments). Data in **h** and **i** are depicted as the mean ± s.d. Significance was assessed using ordinary one-way analysis of variance (ANOVA) with Tukey's multiple comparisons.

ApoE⁺CD11b⁺ AMs. To investigate the β-glucan-induced cellular dynamics of ApoE⁺CD11b⁺ AMs we monitored the BALF from day 0 to day 21 after β-glucan exposure using flow cytometry (Fig. 1h,i and Supplementary Fig. 1h,i). This analysis revealed that, in line with the single-cell transcriptomic data, ApoE⁺CD11b⁺ AMs peaked at day 7 after β-glucan inoculation and gradually declined until day 21 (Fig. 1h,i). In agreement with this, we observed an overall increase in total AMs peaking at day 7 after β-glucan exposure (Supplementary Fig. 1j,k). Generation of

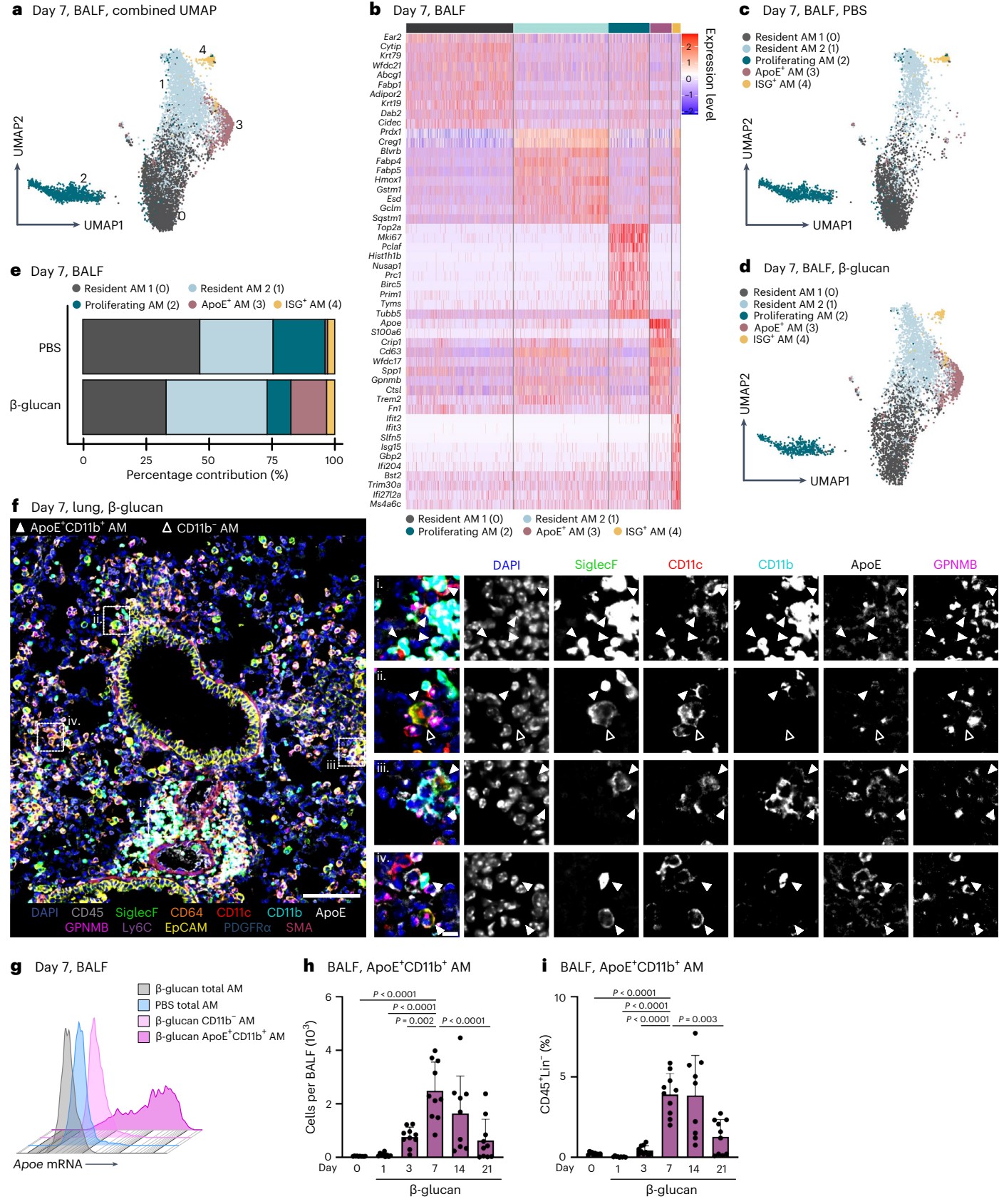

**a** Day 7, BALF, combined UMAP

**b** Day 7, BALF

**c** Day 7, BALF, PBS
- Resident AM 1 (0)
- Resident AM 2 (1)
- Proliferating AM (2)
- ApoE⁺ AM (3)
- ISG⁺ AM (4)

**d** Day 7, BALF, β-glucan
- Resident AM 1 (0)
- Resident AM 2 (1)
- Proliferating AM (2)
- ApoE⁺ AM (3)
- ISG⁺ AM (4)

**e** Day 7, BALF
- Resident AM 1 (0)
- Resident AM 2 (1)
- Proliferating AM (2)
- ApoE⁺ AM (3)
- ISG⁺ AM (4)

Percentage contribution (%)

- Resident AM 1 (0)
- Resident AM 2 (1)
- Proliferating AM (2)
- ApoE⁺ AM (3)
- ISG⁺ AM (4)

**f** Day 7, lung, β-glucan
▲ ApoE⁺CD11b⁺ AM    △ CD11b⁻ AM

DAPI  CD45  SiglecF  CD64  CD11c  CD11b  ApoE
GPNMB  Ly6C  EpCAM  PDGFRα  SMA

DAPI  SiglecF  CD11c  CD11b  ApoE  GPNMB

**g** Day 7, BALF
- β-glucan total AM
- PBS total AM
- β-glucan CD11b⁻ AM
- β-glucan ApoE⁺CD11b⁺ AM

Apoe mRNA

**h** BALF, ApoE⁺CD11b⁺ AM
Cells per BALF (10³)
P < 0.0001
P < 0.0001
P = 0.002   P < 0.0001
Day  0  1  3  7  14  21
β-glucan

**i** BALF, ApoE⁺CD11b⁺ AM
CD45⁺Lin⁻ (%)
P < 0.0001
P < 0.0001
P < 0.0001   P = 0.003
Day  0  1  3  7  14  21
β-glucan

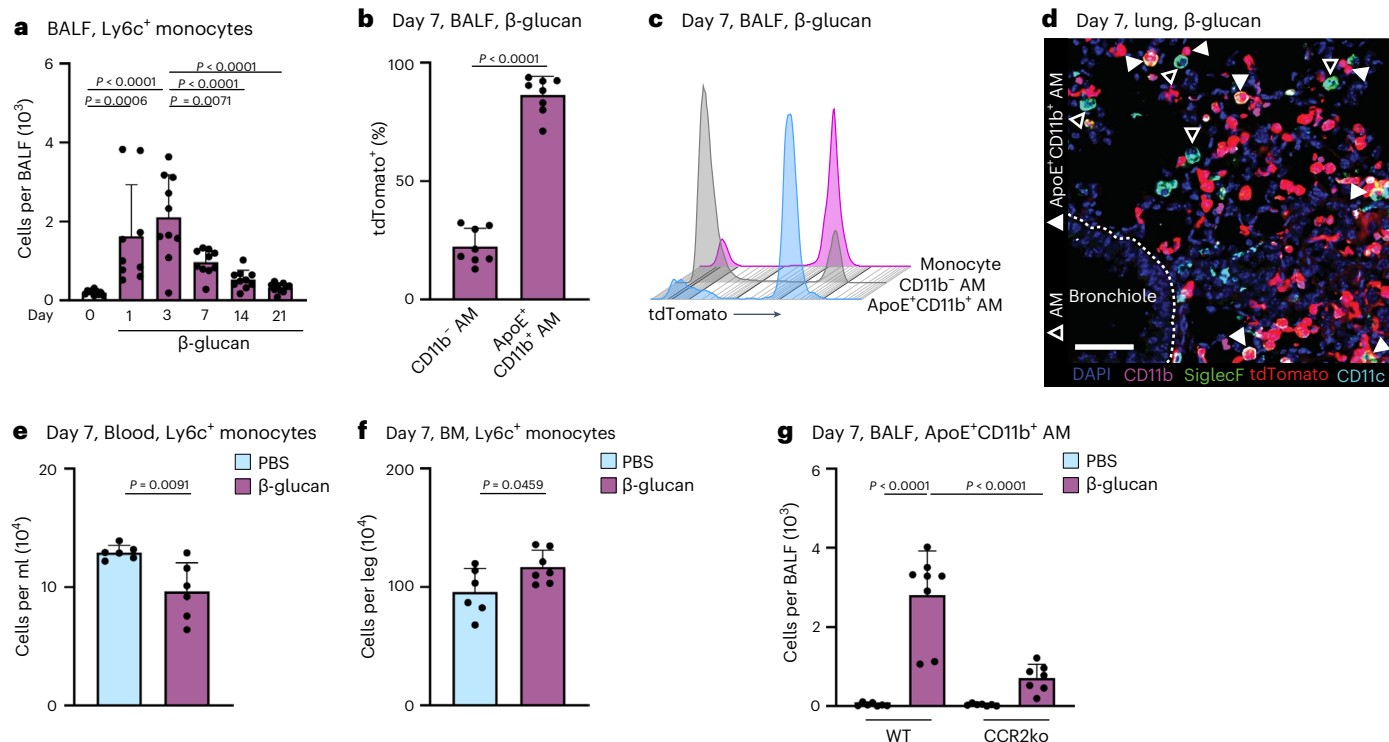

**Fig. 2 | ApoE⁺CD11b⁺ AMs are monocyte derived and CCR2 dependent.**
**a**, Flow cytometric quantification of absolute Ly6c⁺ monocyte (CD45⁺Ly6g⁻Siglec-F⁻CD64^{int}CD11b⁺Ly6c⁺) numbers in the BALF 1 to 21 days after β-glucan stimulation of WT mice ($n$ = 9–10 mice, two independent experiments). **b,c**, Flow cytometric analysis of BALF from *Ms4a3-cre^{Rosa26TOMATO}* mice 7 days after intranasal PBS or β-glucan stimulation ($n$ = 8, two independent experiments). Percentage of tdTomato⁺ labeling in CD11b⁻ and ApoE⁺CD11b⁺ AMs (**b**) and proportion of tdTomato⁺ labeling in CD11b⁻ and ApoE⁺CD11b⁺ AMs compared to monocytes (CD45⁺Siglec-F⁻Ly6g⁻CD11b⁺F4/80⁺) (**c**). **d**, CODEX multiplexed immunostaining of the left lobe of a *Ms4a3-cre^{Rosa26TOMATO}* mouse 7 days after β-glucan exposure (enlargement from Fig. 1f). Filled arrowheads indicate ApoE⁺CD11b⁺ AMs, whereas empty arrowheads indicate CD11b⁻ AMs. tdTomato reporter signals are represented in red. Scale bar, 50 μm. **e,f**, Absolute counts of Ly6c⁺ monocytes in the blood (**e**) or BM (**f**) of WT mice 7 days after PBS or β-glucan by flow cytometry ($n$ = 6 mice, two individual experiments). **g**, Flow cytometric quantification of absolute ApoE⁺CD11b⁺ AM numbers in the BALF 7 days after β-glucan exposure in WT or CCR2⁻/⁻ mice ($n$ = 7–8 mice, two individual experiments). Data are depicted as the mean ± s.d. Significance was assessed using ordinary one-way ANOVA with Tukey's multiple comparisons (**a** and **g**) and unpaired two-tailed student's $t$-test (**b**, **e** and **f**).

ApoE⁺CD11b⁺ AMs was associated with a transient influx of neutrophils and eosinophils on days 1 and 3 (Supplementary Fig. 1l,m). To confirm the macrophage identity of ApoE⁺CD11b⁺ AMs, Siglec-F expression on CD11b⁻ AMs, ApoE⁺CD11b⁺ AMs and Ly6c⁺ monocytes was assessed. Both CD11b⁻ and ApoE⁺CD11b⁺ AMs expressed high levels of Siglec-F, whereas Ly6c⁺ monocytes did not, further establishing ApoE⁺CD11b⁺ AMs as part of the AM compartment (Supplementary Fig. 1n). Finally, to understand the abundance ApoE⁺CD11b⁺ AMs in total lung single-cell suspensions, we quantified ApoE⁺CD11b⁺ AMs using flow cytometry (Supplementary Fig. 1o). ApoE⁺CD11b⁺ AMs were also present in full lung suspensions, displaying similar quantity and marker profiles as in BALF. In summary, intranasal β-glucan induces ApoE⁺CD11b⁺ AMs as a cellular response to environmental stimulation.

### ApoE⁺CD11b⁺ AMs are derived from monocytes and depend on CCR2

Both acute and chronic inflammation induce the recruitment of MoMacs into the bronchoalveolar space[16,31,34]. To understand whether β-glucan-induced environmental adaptation induces a new resident AM cell state or results in the recruitment and differentiation of Ly6c⁺ monocytes into ApoE⁺CD11b⁺ AMs, we tracked the influx of Ly6c⁺ monocytes into the BALF using flow cytometry. After β-glucan stimulation, Ly6c⁺ monocytes were recruited to the bronchoalveolar space, peaking 3 days after stimulation, remaining elevated on day 7, and gradually declining from day 14 onwards (Fig. 2a and

Supplementary Fig. 2a). To connect these findings with the emergence of ApoE⁺CD11b⁺ AMs, we utilized the *Ms4a3-cre^{Rosa26TOMATO}* mice, enabling genetic tracing of BM-derived granulocyte-macrophage progenitors (GMPs).

Genetic lineage tracing revealed that 86% ± 7.8% of ApoE⁺CD11b⁺ AMs were labeled with tdTomato, indicating a BM GMP lineage origin (Fig. 2b,c). CODEX imaging showed coexpression of CD11b and tdTomato in Siglec-F⁺CD11c⁺ AMs within tissue sections from mice stimulated with β-glucan 7 days prior, supporting their GMP and monocyte origin (Fig. 2d). To determine if Ly6c⁺ monocytes are systemically mobilized and recruited to the lung from the BM following β-glucan stimulation, we assessed the abundance of Ly6c⁺ monocytes, common monocyte progenitors (cMOPs) and GMPs in the blood and BM (Fig. 2e,f and Supplementary Fig. 2b–g). This revealed a reduction in Ly6c⁺ monocytes in the blood, accompanied by a compensatory increase in BM Ly6c⁺ monocytes 7 days after β-glucan stimulation. To confirm the monocytic BM origin of ApoE⁺CD11b⁺ AMs, we utilized CCR2-deficient mice, in which the recruitment of Ly6c⁺ monocytes into peripheral tissues is impaired[35]. To investigate dependence of ApoE⁺CD11b⁺ AMs on CCR2, we intranasally inoculated control and CCR2-deficient mice with β-glucan and analyzed the BALF 7 days later using flow cytometry (Fig. 2g and Supplementary Fig. 2h). This analysis revealed a significant reduction in ApoE⁺CD11b⁺ AMs in CCR2-deficient mice following β-glucan inoculation. In summary, these results collectively demonstrate that ApoE⁺CD11b⁺ AMs induced by β-glucan exposure originate from BM Ly6c⁺ monocytes in a CCR2-dependent manner.

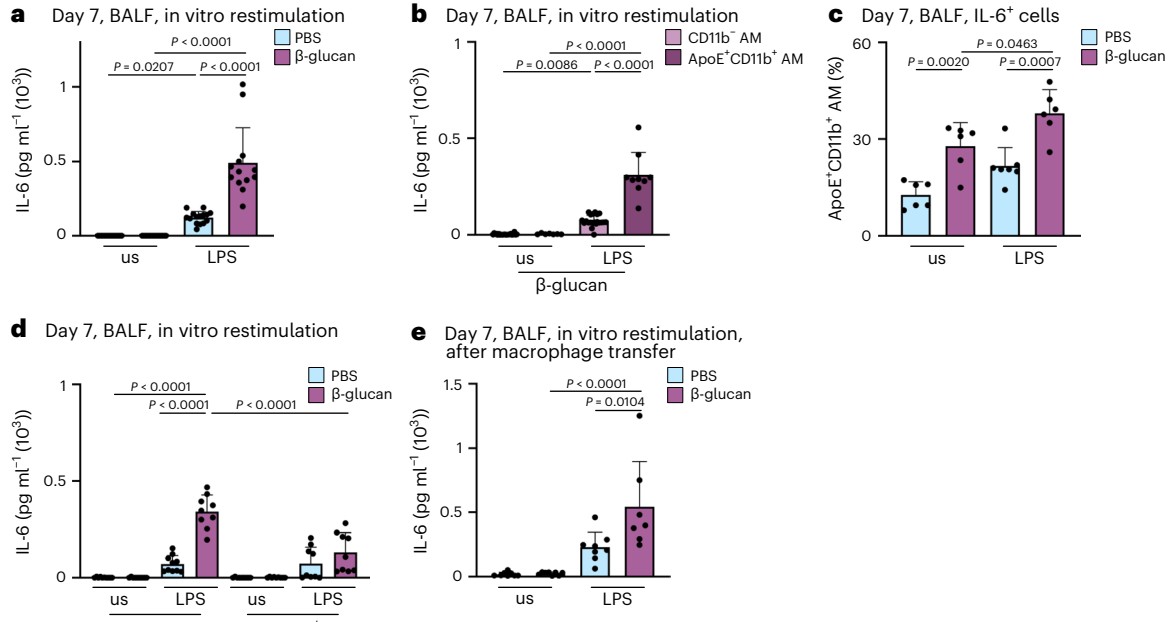

**Fig. 3 | ApoE⁺CD11b⁺ AMs show increased release of IL-6 and induction of glycolysis. a–c**, BALF cells were harvested from WT or *Ms4a3-cre^ROSA26TOMATO* mice 7 days after intranasal exposure with PBS or β-glucan and subsequently restimulated in vitro with or without LPS (unstimulated, us) for 24 h. **a**, Quantification of IL-6 protein levels by ELISA in the cell culture supernatant 24 h after LPS restimulation of WT mice (*n* = 13–15 mice, three individual experiments). **b**, CD11b⁺Ms4a3⁺ AMs and CD11b⁻Ms4a3⁻ AMs were sorted from the pooled BALF of PBS or β-glucan-stimulated *Ms4a3-cre^ROSA26TOMATO* mice and seeded with 0.2 × 10⁵ cells per well before LPS restimulation (*n* = 9 mice for PBS, *n* = 31 mice for β-glucan; one dot represents the pooled supernatant of two technical replicate wells, minimum of nine data points per group, two individual experiments). **c**, Percentage of IL-6⁺ cells among ApoE⁺CD11b⁺

AMs after restimulation with LPS for 6–8 h followed by intracellular staining and flow cytometric analysis (*n* = 6–7 mice, two individual experiments). **d**, Quantification of IL-6 protein levels by ELISA in the cell culture supernatant 24 h after restimulation with LPS of WT and CCR2⁻/⁻ mice (*n* = 9–10 mice, two individual experiments). **e**, BALFs of PBS or β-glucan-experienced CD45.2 WT mice were harvested and pooled 5 days after stimulation. 2 × 10⁵ cells in 35 µl were intratracheally transferred into CD45.1 mice. Quantification of IL-6 protein levels by ELISA in the cell culture supernatant after restimulation with LPS 48 h after transfer (*n* = 7–8 mice, two individual experiments). Data are depicted as the mean ± s.d. Significance was assessed using ordinary one-way ANOVA with Tukey's multiple comparisons (**a–e**).

## ApoE⁺CD11b⁺ AMs exhibit an elevated release of IL-6

MoMacs have been linked to heightened inflammatory responses following high-grade inflammatory and infectious events, such as influenza A virus infection[16]. Specifically, the increased production of IL-6 is a hallmark feature of functionally modified MoMacs during acute inflammation. However, it remains unclear whether BALF-resident mononuclear cells functionally adapt to local low-grade inflammation. To address this, we investigated the functional profile of AMs 7 days after exposure to β-glucan. We isolated BALF AMs, and subsequently restimulated them with PBS or lipopolysaccharide (LPS) for 24 h in vitro. Analysis of IL-6 release in the supernatants using ELISA revealed that macrophages pre-exposed in vivo to β-glucan released significantly higher amounts of IL-6 upon LPS restimulation, compared to their PBS-pretreated counterparts (Fig. 3a). To identify the cellular source responsible for increased IL-6 production, we purified BALF-resident CD11b⁻ and ApoE⁺CD11b⁺ AMs of intranasal β-glucan-treated mice 7 days earlier and restimulated them with LPS (Fig. 3b). Only ApoE⁺CD11b⁺ AMs released comparable amounts of IL-6 to those observed in complete BALF AM preparations (Fig. 3a). Subsequently, intracellular flow cytometric analysis demonstrated a significant increase in IL-6⁺CD11b⁺ AMs following β-glucan exposure compared to PBS-exposed controls (Fig. 3c and Supplementary Fig. 3a). Our data indicate that the generation of ApoE⁺CD11b⁺ AMs in response to β-glucan exposure depends on CCR2 (Fig. 2g). To confirm that the increased IL-6 observed in ex vivo restimulated AMs can be directly attributed to CCR2-dependent ApoE⁺CD11b⁺ AMs, we exposed CCR2-deficient and control mice to β-glucan. Seven days later, BALF AMs were enriched, restimulated and restimulated with LPS for 24 h in vitro (Fig. 3d). This analysis revealed

that the elevated IL-6 levels observed in β-glucan-exposed BALF AMs are CCR2 dependent, providing evidence that ApoE⁺CD11b⁺ AMs are the primary source of increased IL-6 during β-glucan-induced environmental adaptation. Finally, to causally establish whether the enhanced IL-6 production is an intrinsic cellular feature of ApoE⁺CD11b⁺ AMs, we transferred CD45.2⁺ BALF-resident AMs into naive CD45.1⁺ mice 5 days after β-glucan-induced environmental adaptation. Two days later, we restimulated BALF AMs with LPS in vitro (Fig. 3e and Supplementary Fig. 3b,c). Transfer of β-glucan-experienced BALF AMs led to an increased IL-6 production upon in vitro restimulation of AMs within the recipient mouse. These findings causally establish the intrinsic β-glucan-induced functional change in ApoE⁺CD11b⁺ AMs.

## β-glucan aids lung bacterial defense and experimental fibrosis recovery

Previous studies associated altered cytokine responses following systemic β-glucan stimulation with increased glycolysis in MPs[7,8]. To investigate this in our system, we measured glycolysis. This revealed a significant increase in glycolysis, glycolytic capacity and glycolytic reserve 7 days after β-glucan exposure in BALF AMs (Fig. 4a,b and Supplementary Fig. 4a,b). In addition, induction of glycolysis in MPs is associated with enhanced phagocytosis[36,37]. Thus, we assessed BALF AM phagocytosis of *Staphylococcus aureus*-coated particles in vitro isolated from mice adapted to PBS or β-glucan 7 days prior. This demonstrated a significant increase in phagocytic activity in BALF AMs isolated from β-glucan-adapted mice but not PBS-adapted mice (Fig. 4c,d and Supplementary Fig. 4c,d). This prompted us to investigate the in vivo functional impact of intranasal β-glucan adaptation in response to

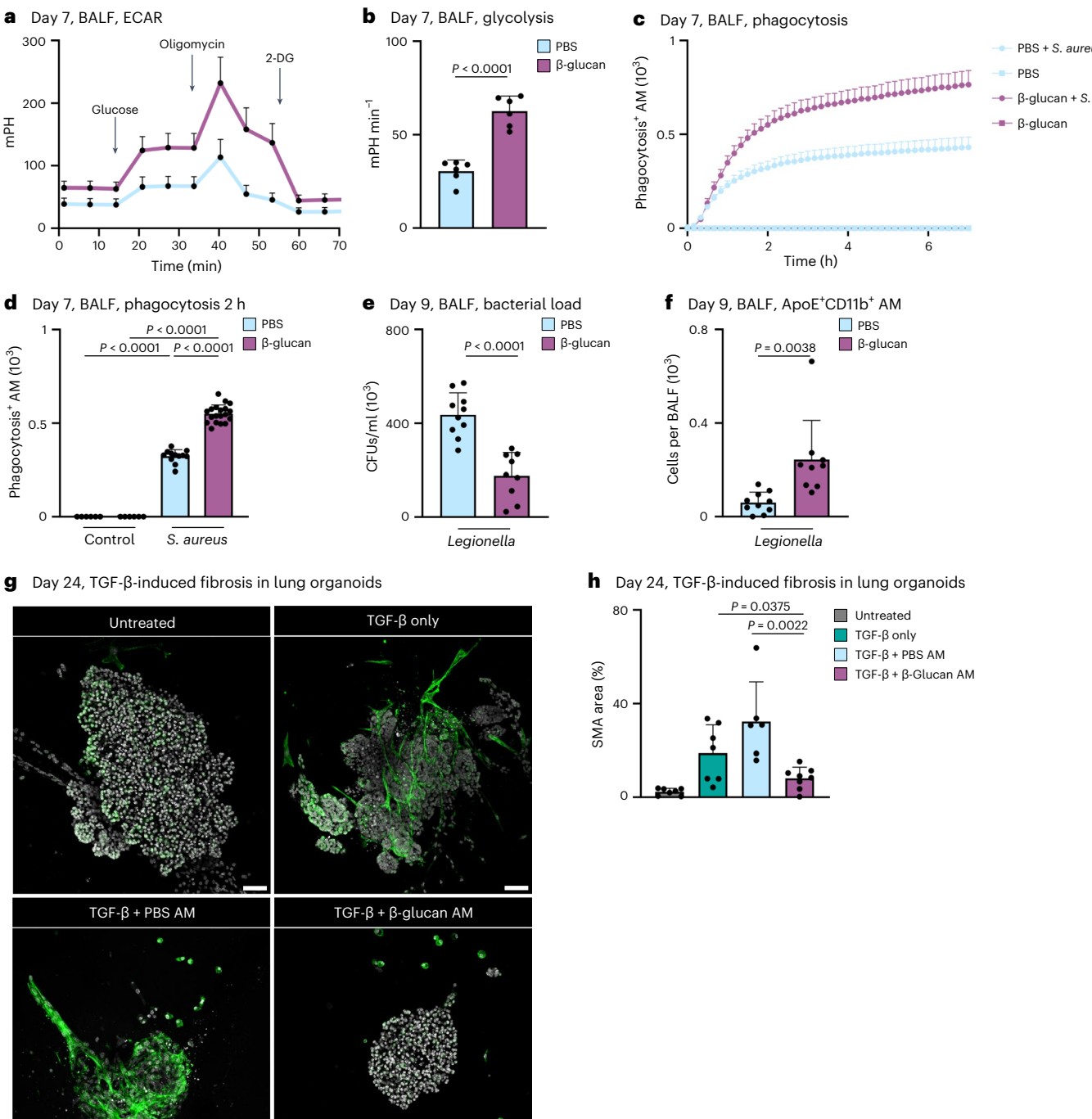

**Fig. 4 | Environmental adaptation induced by β-glucan significantly improves pulmonary bacterial clearance and outcome of bleomycin-induced fibrosis. a,b**, Extracellular acidification rate (ECAR; **a**) and glycolysis (**b**) in BALF cells 7 days after PBS or β-glucan stimulation of WT mice measured by Seahorse (*n* = 6 mice, one independent experiment). **c,d**, AM cells from day 7 PBS- or β-glucan-experienced WT mice were selected by adherence and subsequently treated with 2.5 μg pHrodo *S. aureus* bioparticles (*n* = 3 mice pooled per condition, technical replicates: 6 control wells, 12–19 treated wells per group, one of two independent experiments shown). **c**, Representative curve of absolute phagocytosis⁺ AM numbers over the time course of 7 h (here shown as mean ± s.d. of all technical replicates). **d**, Absolute numbers of phagocytosis⁺ AMs 2 h after adding the pHrodo *S. aureus* bioparticles. **e,f**, C57BL/6J WT mice were intranasally stimulated with PBS or β-glucan followed by intratracheal infection with 5 × 10⁶ colony-forming units (CFUs) *L. pneumophilia* at day 7 after

primary stimulation and analysis at day nine (*n* = 9–10 mice, two independent experiments). Quantification of bacterial load in BALF (**e**) and absolute numbers of ApoE⁺CD11b⁺ AMs by flow cytometry 9 days after primary stimulation (**f**). **g,h**, Representative confocal images (**g**) and SMA area quantification (**h**) of BALOs co-cultured with PBS- or β-glucan-experienced AMs 48 h after induction of fibrosis via TGF-β. Seven days after stimulation, 2.5 × 10⁴ AMs of PBS- or β-glucan-experienced WT mice were co-cultured with day 21 lung BALOs for 24 h. AM–organoid co-cultures were subsequently treated with 1.05 ng ml⁻¹ TGF-β for 48 h before fixation and antibody staining. Myofibroblasts were stained for α-SMA (*n* = 6–8 organoids per condition from two replicate wells; one of two independent experiments shown). Scale bars, 50 μm (**g**). Data are depicted as the mean ± s.d. Significance was assessed using unpaired two-tailed student's *t*-test (**b**, **e** and **f**), ordinary one-way ANOVA with Tukey's multiple comparisons (**d**) and two-tailed Mann–Whitney test (**h**).

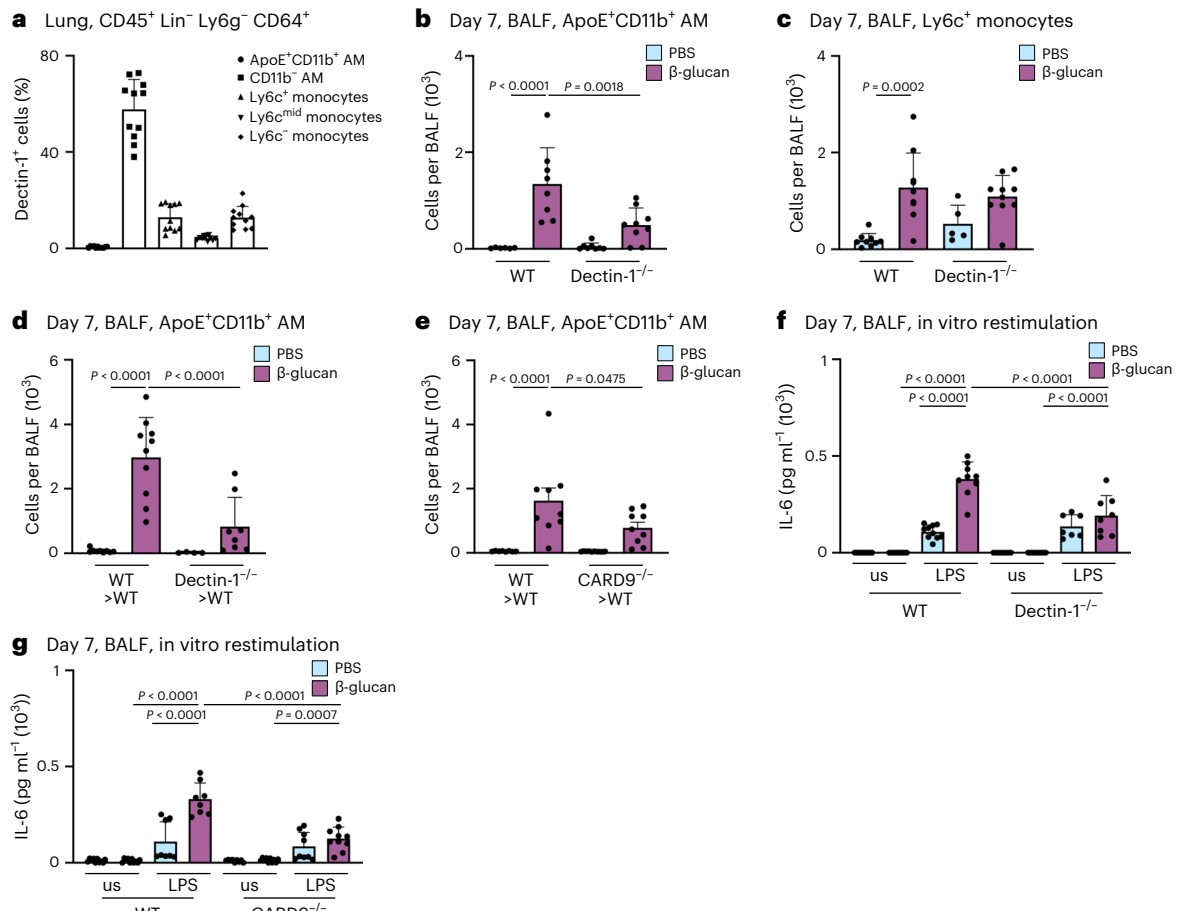

**Fig. 5 | Generation of ApoE⁺CD11b⁺ AMs by β-glucan is dependent on the Dectin-1–CARD9 signaling axis. a**, Percentage of monocyte and macrophage populations contributing to Dectin-1⁺ cells in the WT mouse lung pregated on CD45⁺Lin⁻Ly6g⁻CD64⁺ cells ($n = 11$ mice, two independent experiments) by flow cytometry. **b,c**, Absolute ApoE⁺CD11b⁺ AM (**b**) and Ly6c⁺ monocyte (**c**) numbers in the BALF 7 days after PBS or β-glucan exposure in WT or *Dectin*1⁻/⁻ mice ($n = 5–9$, two independent experiments) by flow cytometry. **d,e**, Absolute ApoE⁺CD11b⁺ AM numbers in the BALF 7 days after PBS or β-glucan exposure in *Dectin1*⁻/⁻ (**d**; $n = 4–10$ mice, two independent experiments) or *Card9*⁻/⁻ (**e**; $n = 8–9$ mice, two

independent experiments) BM chimeras by flow cytometry. **f**, Quantification of IL-6 protein levels by ELISA in the cell culture supernatant 24 h after LPS restimulation of WT and *Dectin1*⁻/⁻ mice ($n = 7–10$ mice, two independent experiments). **g**, Quantification of IL-6 protein levels by ELISA in the cell culture supernatant 24 h after LPS restimulation of WT and *Card9*⁻/⁻ mice ($n = 9–10$ mice, two independent experiments). Data are depicted as the mean ± s.d. Significance was assessed using ordinary one-way ANOVA with Tukey's multiple comparisons (**b–g**).

acute bacterial challenge. We infected C57BL/6 mice that were β-glucan or PBS adapted 7 days earlier with *L. pneumophila* and analyzed the BALF's bacterial burden and cellular composition 2 days after infection[38]. β-glucan-adapted mice exhibited a significant reduction in bacteria detected in BALF, along with an increased count of pro-inflammatory macrophages associated with bacterial clearance (Fig. 4e,f and Supplementary Fig. 4e).

Furthermore, we examined whether β-glucan adaptation had effects beyond the modulation of acute bacterial infection using bleomycin-induced experimental lung fibrosis. β-glucan-adapted mice showed significantly higher survival rates, lower disease burden and reduced weight loss over a 14-day observation period after bleomycin inoculation (Supplementary Fig. 4f–i). Moreover, the pro-resolution associated effectors, IL-4 and IL-33, were enhanced on day 3 after bleomycin inoculation, while on day 14 after bleomycin thymic stromal lymphopoietin (TSLP) decreased in β-glucan-adapted mice (Supplementary Fig. 4j–l). No difference in lung fibrotic area was observed (Supplementary Fig. 4m,n). These findings highlight the substantial regulatory role of ApoE⁺CD11b⁺ AMs elicited by environmental adaptation in the control and severity of acute and chronic inflammation. To elucidate the direct effect of ApoE⁺CD11b⁺ AMs on the development of

lung fibrosis, we generated bronchoalveolar lung organoids (BALOs) containing myofibroblasts[39]. We treated them with transforming growth factor-beta (TGF-β) to induce a fibrotic response. BALF AMs isolated from β-glucan or PBS-adapted mice were added to day 21 BALOs 24 h before TGF-β pro-fibrotic stimulation and co-cultured for 48 h. Adding TGF-β to BALOs led to the increased production of fibroblast smooth muscle actin (SMA), a hallmark of lung fibrosis. We quantified SMA production in β-glucan or PBS-adapted AM-supplemented fibrotic BALOs. We observed a significant reduction in SMA production when β-glucan-adapted AMs were added, while the addition of PBS-adapted AMs showed no effect (Fig. 4g,h and Supplementary Fig. 4o). In conclusion, functional in vivo and in vitro data establish ApoE⁺CD11b⁺ AMs as crucial environmentally induced modulators of lung inflammation, providing valuable insights into the molecular mechanisms underlying their role in mitigating fibrosis.

### β-glucan induces ApoE⁺CD11b⁺ AMs via Dectin-1/CARD9
β-glucan is recognized by various receptors, including CR3, Dectin-1 and CD5 (refs. 40–42). Dectin-1 is most prominently expressed on MPs. To understand how Dectin-1 regulates ApoE⁺CD11b⁺ AMs, we used flow cytometry to profile its expression on BALF macrophages (Fig. 5a). This

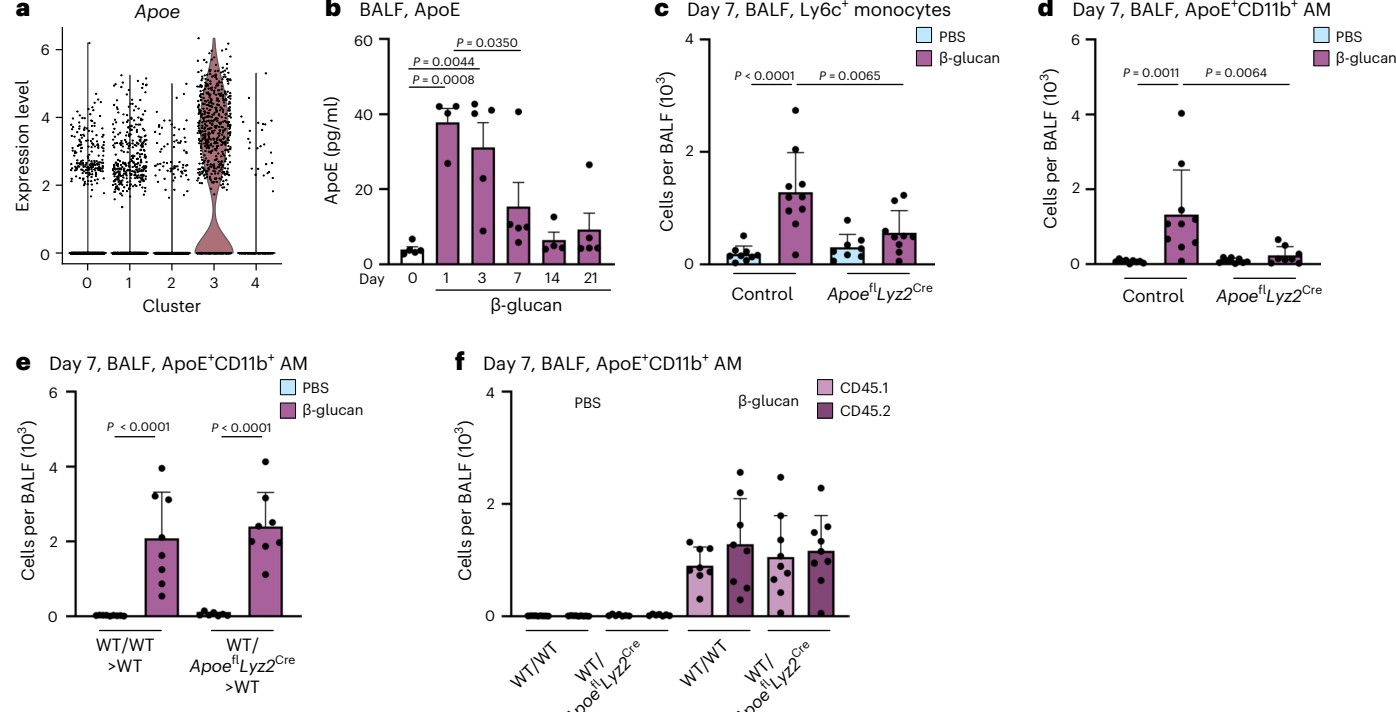

**Fig. 6 | Paracrine myeloid-derived ApoE controls ApoE⁺CD11b⁺ AM differentiation upon β-glucan-induced environmental adaptation. a**, Violin plot of *Apoe* RNA expression levels in the BALF 7 days after β-glucan exposure by scRNA-seq. **b**, WT mice were stimulated with β-glucan, and BALF was harvested at different time points. The plot shows ApoE protein levels in the BALF measured by ELISA (*n* = 4–5 mice, one independent experiment). **c,d**, Absolute numbers of ApoE⁺CD11b⁺ AM (**c**) and Ly6c⁺ monocytes (**d**) 7 days after intranasal β-glucan exposure of control or *Apoe*ᶠˡ*Lyz2*ᶜʳᵉ mice by flow cytometry (*n* = 8–10 mice, three independent experiments). **e,f**, Lethally irradiated CD45.1⁺/CD45.2⁺ male mice

were reconstituted with 1.5 × 10⁶ CD45.1⁺ mixed with CD45.2⁺ BM cells (WT/WT) or with CD45.1⁺ mixed with *Apoe*ᶠˡ*Lys2*ᶜʳᵉ CD45.2⁺ BM cells (WT/*Apoe*ᶠˡ*Lyz2*ᶜʳᵉ) for 12 weeks and subsequently intranasally stimulated with PBS or β-glucan (*n* = 8–9 mice, two independent experiments). Flow cytometric quantification of ApoE⁺CD11b⁺ AM numbers (**e**) and contribution of donor cells (CD45.1⁺ or CD45.2⁺) to the ApoE⁺CD11b⁺ AM pool (**f**) 7 days after exposure. Data are depicted as the mean ± s.d. Significance was assessed using ordinary one-way ANOVA with Tukey's multiple comparisons (**b–f**).

revealed that homeostatic Dectin-1 expression is largely confined to resident AMs with only a small fraction of monocytes expressing Dectin-1. To assess the role of Dectin-1 for the development of ApoE⁺CD11b⁺ AMs, we intranasally inoculated control or Dectin-1⁻/⁻ mice with β-glucan and used flow cytometry to analyze BALF-resident immune cells 7 days later. This revealed that generation of ApoE⁺CD11b⁺ AMs is dependent on Dectin-1 expression, whereas initial inflammatory recruitment of Ly6c⁺ monocytes to the BALF is not (Fig. 5b,c). Next, to understand whether immune cell-intrinsic or stromal cell recognition via Dectin-1 is critical for the development of ApoE⁺CD11b⁺ AMs and thus environmental adaptation, we transferred Dectin-1⁻/⁻ or control (CD45.2⁺) BM into lethally irradiated CD45.1⁺ control mice and analyzed their BALF 7 days after environmental adaptation by β-glucan. Here, generation of ApoE⁺CD11b⁺ AMs was entirely dependent on hematopoietic expression of Dectin-1 (Fig. 5d and Supplementary Fig. 5a,b). CARD9 mediates activation of NF-κB by Dectin-1 (refs. 12,13,43). To investigate if ApoE⁺CD11b⁺ AMs require CARD9 for their development, we treated lethally irradiated mice reconstituted with *Card9*⁻/⁻ or control BM with β-glucan or PBS and flow cytometrically analyzed the BALF 7 days later. This revealed that development of ApoE⁺CD11b⁺ AMs depends on Dectin-1-elicited CARD9-dependent signaling (Fig. 5e and Supplementary Fig. 5c,d). Next, we investigated whether the loss of ApoE⁺CD11b⁺ AMs by abrogating Dectin-1 or CARD9 signaling leads to a loss of increased IL-6 secretion upon in vitro LPS restimulation in BALF macrophages. In line with the data obtained in the CCR2⁻/⁻ mouse model, enhanced IL-6 secretion was abolished in the absence of Dectin-1 or CARD9 signaling and thus can be attributed to ApoE⁺CD11b⁺ AMs (Fig. 5f,g).

### β-glucan triggers ApoE⁺CD11b⁺ AM differentiation via myeloid ApoE

ApoE is expressed in various MoMac populations associated with different low-grade or chronic inflammatory diseases but its role in monocyte-to-macrophage differentiation and maintenance remains unexplored[16,29]. During β-glucan-induced environmental adaptation, ApoE was highly expressed in ApoE⁺CD11b⁺ AMs and detectable at the protein level as early as 1 day after intranasal β-glucan stimulation, coinciding with BALF Ly6c⁺ monocyte recruitment (Figs. 2a and 6a,b). To elucidate the role of ApoE in the environmental adaptation of the lung MP repertoire, we intranasally inoculated *Apoe*ᶠˡ*Lyz2*ᶜʳᵉ mice, which lack ApoE expression within the myeloid lineage, with β-glucan. Next, we used flow cytometry to analyze the composition of the BALF MP compartment 7 days later. β-glucan-stimulated *Apoe*ᶠˡ*Lyz2*ᶜʳᵉ mice did not exhibit increased numbers of BALF Ly6c⁺ monocytes and, as a consequence, failed to generate ApoE⁺CD11b⁺ AMs (Fig. 6c,d and Supplementary Fig. 6a,b).

Next, we examined whether loss of myeloid ApoE influences the observed feedback on blood and BM Ly6c⁺ monocytes. Myeloid ApoE deficiency abrogated the decrease in blood monocytes and the compensatory increase in BM Ly6c⁺ monocytes observed in control mice. Myeloid ApoE deficiency showed no effect on cMOPs or GMPs 7 days after β-glucan stimulation (Supplementary Fig. 6c–f). Subsequently, we investigated whether ApoE controls the generation of ApoE⁺CD11b⁺ AMs through paracrine or autocrine signaling, as both modes have been described previously[44,45]. Here, we generated mixed BM chimeras with a 50:50 ratio of wild-type (WT; CD45.1⁺) and *Apoe*ᶠˡ*Lyz2*ᶜʳᵉ (CD45.2⁺) cells

or congenic mixed WT (CD45.1⁺):WT (CD45.2⁺) control chimeras. After reconstitution, we intranasally stimulated them with β-glucan and used flow cytometry to analyze their BALF MP repertoire 7 days later. This revealed that both WT/WT and WT/*Apoe*ᶠˡ*Lyz2*^Cre chimeras efficiently generated ApoE⁺CD11b⁺ AMs 7 days after β-glucan exposure, supporting a paracrine signaling mode (Fig. 6e and Supplementary Fig. 6g,h). Next, we examined the contribution of ApoE-deficient CD45.2⁺ cells to the pool of total ApoE⁺CD11b⁺ AMs. We found that both ApoE-proficient (CD45.1) and ApoE-deficient (CD45.2) cells equally contributed to the pool of β-glucan-stimulated ApoE⁺CD11b⁺ AMs (Fig. 6f). This demonstrates that a paracrine myeloid cell-derived source of ApoE is sufficient to rescue the generation of ApoE⁺CD11b⁺ AMs during environmental adaptation in the lung.

## Myeloid-derived ApoE controls survival of ApoE⁺CD11b⁺ AMs by regulation of cholesterol storage and M-CSF secretion

To elucidate ApoE's role in the differentiation and survival of ApoE⁺CD11b⁺ AMs, we conducted experiments to determine whether myeloid-derived ApoE influences the initial commitment of Ly6c⁺ monocytes to or the maintenance and survival of MoMacs. We used control and *Apoe*ᶠˡ*Lyz2*^Cre mice 3 days after intranasal β-glucan exposure. On day 3 after β-glucan exposure, similar numbers of BALF Ly6c⁺ monocytes were present in control and *Apoe*ᶠˡ*Lyz2*^Cre mice. This suggests that ApoE does not regulate the initial commitment to the macrophage lineage or the recruitment of monocytic precursors to the BALF. However, our results established a critical time window during which ApoE is essential for monocyte-to-macrophage differentiation following β-glucan inoculation (Fig. 7a,b and Supplementary Fig. 7a,b). To investigate potential molecular dysregulation caused by the absence of ApoE in differentiating macrophages, we assessed intracellular cholesterol content and distribution of BALF ApoE⁺CD11b⁺ AMs 3 days after β-glucan stimulation. This revealed that ApoE-deficient CD11b⁺ AMs have increased intracellular cholesterol, as indicated by filipin staining (Fig. 7c,d). Previous data linked dysregulated ApoE signaling to cholesterol accumulation in the endoplasmic reticulum and a reduction in protein synthesis[46]. To test these mechanisms in our experimental system of monocyte-to-macrophage differentiation, we analyzed the colocalization of BODIPY-cholesterol with the endoplasmic reticulum-associated protein calreticulin and early or late endosome markers (EEA1 and LAMP1) using confocal microscopy. This revealed that, in ApoE-deficient differentiating macrophages 3 days after β-glucan exposure, BODIPY-cholesterol colocalizes with calreticulin, accumulating at the endoplasmic reticulum (Fig. 7e,f). M-CSF-releasing monocytes differentiating into macrophages have been described to be crucial for lung monocyte-to-macrophage differentiation[47]. To assess whether aberrant cholesterol accumulation affects the M-CSF–M-CSF

receptor (CSF-1R) macrophage survival circuit, we monitored BALF and lung tissue intracellular and extracellular M-CSF secretion. Here, ApoE deficiency resulted in a reduction of both BALF and lung tissue extracellular and intracellular production of M-CSF on day 1 and 3 after β-glucan exposure (Fig. 7g,h and Supplementary Fig. 7c–e), leading to an increase in TUNEL⁺ ApoE-deficient CD11b⁺ AMs in *Apoe*ᶠˡ*Lyz2*^Cre mice (Fig. 7i). To determine if the loss of M-CSF plays a crucial molecular role in the loss of differentiating monocytes to macrophages following β-glucan-induced environmental adaptation, we used antibody-mediated CSF-1R blockade on days 0 and 3 after β-glucan stimulation and analyzed treated and control animals 7 days later. Mice treated with anti-CSF-1R-blocking antibody exhibited significantly lower numbers of ApoE⁺CD11b⁺ AMs 7 days after β-glucan inoculation, underscoring the importance of M-CSF in the monocyte-to-macrophage differentiation process following β-glucan stimulation (Fig. 7j and Supplementary Fig. 7f–i). Taken together, this suggests ApoE as a central regulator of pulmonary monocyte-to-macrophage differentiation and survival via the M-CSF signaling axis upon β-glucan-induced environmental adaptation.

## Discussion

Within our modern-day environment, the lung is constantly exposed to a plethora of sterile immunostimulatory components. However, the developmental, functional and molecular consequences for lung-resident macrophages are incompletely understood. Here, we show that a single non-pathological intranasal β-glucan stimulus induces the development of MoAMs, which highly express CD11b and ApoE and are characterized by their superior IL-6 production capacity in response to secondary LPS stimulation. Additionally, ApoE⁺CD11b⁺ AMs are glycolytic, highly phagocytic and modify the outcome of a secondary bacterial infection and of a chronic fibrotic response in vivo. Molecularly, this is instructed by Dectin-1-mediated recognition of β-glucan and its signaling adaptor protein CARD9. Further analysis revealed a crucial role of ApoE for the maintenance of BALF-resident ApoE⁺CD11b⁺ AMs via the control of macrophage-derived M-CSF. This reveals that ApoE is a crucial checkpoint for monocyte-to-macrophage differentiation in the face of environmental adaptation and couples cellular cholesterol metabolism to differentiation and function.

Prior studies examined the development of MoAMs during viral, bacterial and fungal infections, radiation or bleomycin-induced fibrosis[16,18,38,48–51]. However, how bronchoalveolar macrophages adapt their transcriptome, metabolism and function to ambient immunostimulatory components beyond the effects of the acute recognition of such stimuli remains poorly understood[52]. Here, we show that although the initial pulmonary inflammation evoked by β-glucan is minimal, functionally modified MoAMs arise from Ly6c⁺ monocytes within the BALF, a process previously affiliated to viral or bacterial infection, for

---

**Fig. 7 | Myeloid-derived ApoE controls survival of ApoE⁺CD11b⁺ AMs by regulation of cholesterol storage and M-CSF secretion. a,b,** Absolute numbers of ApoE⁺CD11b⁺ AMs (**a**) and Ly6c⁺ monocytes (**b**) 3 days after intranasal β-glucan exposure in control or *Apoe*ᶠˡ*Lyz2*^Cre mice (*n* = 7–8, two independent experiments) by flow cytometry. **c,d,** BALF of PBS or β-glucan-stimulated control or *Apoe*ᶠˡ*Lyz2*^Cre mice was harvested 3 days after exposure, seeded and fixed after 2 h. Filipin staining was performed followed by immunofluorescence analysis. Representative images are shown in **c**. Scale bars, 5 μm. Plot in **d** shows mean filipin signal intensities of individual ApoE⁺CD11b⁺ AMs in the different conditions (*n* = 3 mice per condition, two independent experiments). **e,f,** Three days after stimulation of PBS- or β-glucan-stimulated control or *Apoe*ᶠˡ*LysM*^Cre mice, AMs from the BALF were selected by adherence and afterwards incubated with 0.5 μM BODIPY-cholesterol overnight. Cells were fixed for 15 min the next day and immunofluorescence was performed. **e,** Representative confocal images. Scale bar, 5 μm. **f,** Quantification of overlapping signals of BODIPY-cholesterol and the organelle markers calreticulin (endoplasmic reticulum), EEA1 (endosomes) and LAMP1 (lysosomes) (*n* = 2 mice per condition, two independent experiments). **g,** BALF of PBS- or β-glucan-stimulated control or *Apoe*ᶠˡ*Lyz2*^Cre mice was harvested

24 h after exposure and seeded. Cells were fixed after 2 h and immunostained to detect Siglec-F, CD11b and M-CSF. Plot shows mean M-CSF signal intensities of individual ApoE⁺CD11b⁺ AMs (*n* = 3 mice per condition, two independent experiments). **h,** Quantification of M-CSF protein levels in the BALF 1 day after β-glucan exposure in control or *Apoe*ᶠˡ*Lyz2*^Cre mice (*n* = 6 mice, two independent experiments) measured by ELISA. **i,** BALF of PBS- or β-glucan-stimulated control or *Apoe*ᶠˡ*Lyz2*^Cre mice was harvested 3 days after exposure, seeded and fixed after 2 h. TUNEL staining was performed, followed by conventional immunofluorescence to detect Siglec-F and CD11b. Plot shows mean TUNEL signal intensities of individual ApoE⁺CD11b⁺ AMs (*n* = 3 mice per condition, two independent experiments). **j,** Absolute numbers of ApoE⁺CD11b⁺ AMs in the BALF of WT mice 7 days after intranasal β-glucan treatment together with 500 μg of CSF-1R antibody or the respective isotype control and follow-up treatment 12 h and 3 days later (*n* = 8–9 mice, two independent experiments). Data are depicted as the mean ± s.d. Significance was assessed using ordinary one-way ANOVA with Tukey's multiple comparisons (**a**, **b** and **g–i**), two-tailed Mann–Whitney test (**d** and **f**) and unpaired two-tailed student's *t*-test (**j**). a.u., arbitrary units.

example. This validates the model as suitable for low-grade environmentally induced inflammation. ApoE⁺CD11b⁺ AMs are induced for up to 21 days after β-glucan stimulation leading to functional modification of the macrophage repertoire in the lung, demonstrating the importance of low-grade inflammatory sterile insults in shaping the overall immune competence of the lung-resident macrophage repertoire.

β-glucan, a cell wall component of many pathological and non-pathological fungi, can be found within ambient air[25]. During a fungal infection, monocytes and MoMacs are major antifungal effectors, via ROS and the activation of antifungal neutrophils, in mice and man[50,53,54]. Furthermore, during the later stages of fungal pathogenesis, monocyte descendants are important for induction of CD4⁺ T cell responses[51].

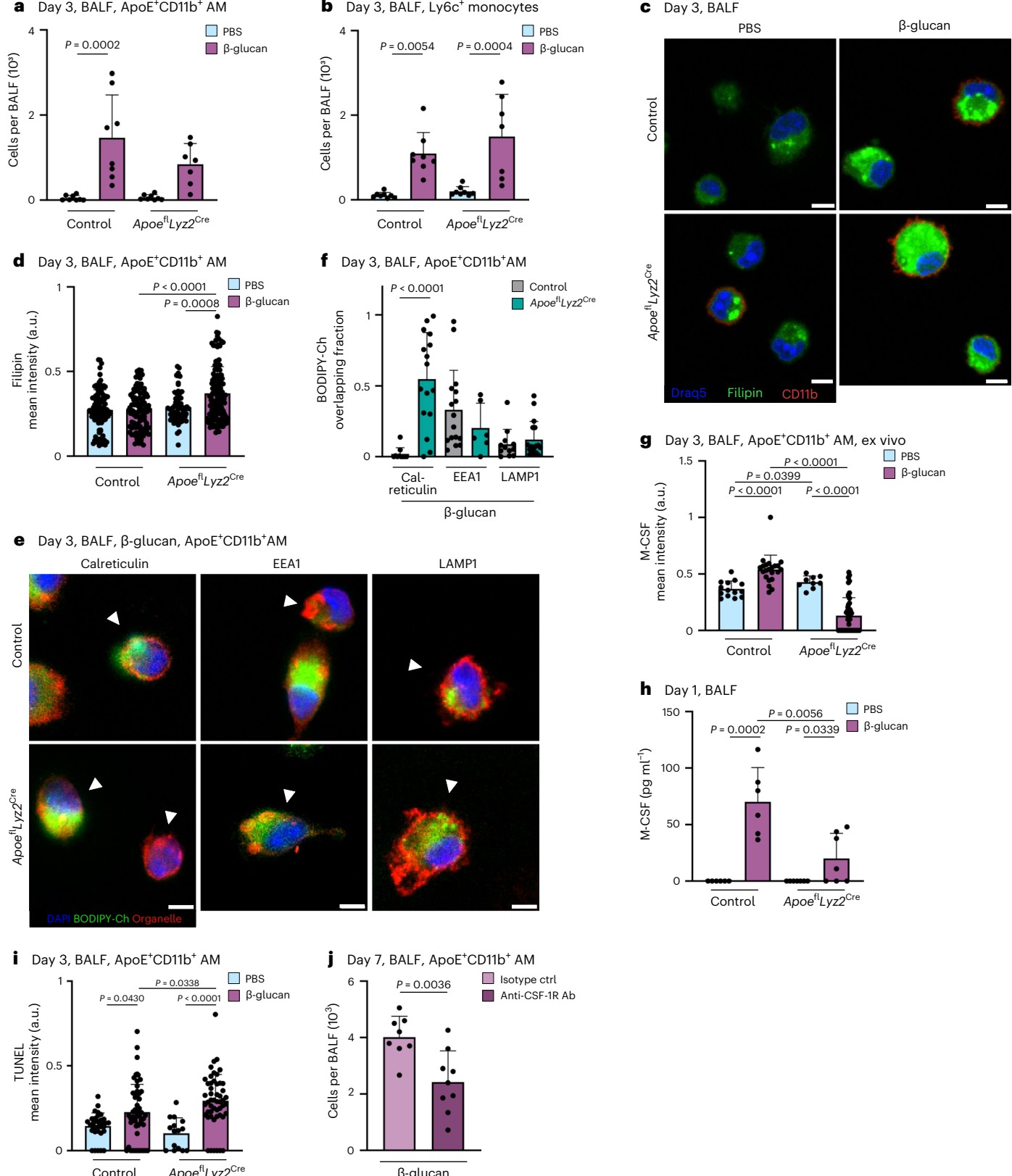

Recently, studies evaluated the role of β-glucan for the induction of systemic innate immune training but did not examine its effects at the level of the tissue[4,55]. Systemically, β-glucan-induced functional modulations were accompanied by the induction of glycolysis and the enhanced release of the pro-inflammatory cytokines IL-6 and TNF in circulating Ly6c⁺ monocytes, similarly to β-glucan-induced functional adaptation in the lung. Systemic β-glucan administration expands BM GMP/multipotent progenitor-like progenitors, resulting in enhanced pathogen clearance or tissue maladaptation[56-58]. We show that pulmonary β-glucan administration only minimally affects BM progenitors and that generation of ApoE⁺CD11b⁺ AMs induces cMOP expansion and recruitment of CCR2-dependent Ly6c⁺ monocytes to the lung. Dectin-1 recognizes β-glucan upon challenge; Dectin-1 downstream signaling is heterogeneous and determines the functional output. We show that tissue adaptation induces Dectin-1 CARD9-dependent signaling circuits, in concordance with increased IL-6 levels, most probably via activation of NF-κB signaling. Finally, β-glucan-induced macrophages upregulate ApoE, a protein demonstrated to be part of various disease-specific MoMac gene signatures, for example, during influenza infection, lung fibrosis or obesity[16,29,48]. Its functional role in MoMac development was not investigated. In hematopoietic stem cells, ApoE was shown to inhibit proliferation and subsequent progenitor maturation by controlling sensitivity towards granulocyte-macrophage stimulating factor and IL-3 (ref. 59). We show that myeloid-specific deletion of ApoE leads to the accumulation of cholesterol at the endoplasmic reticulum, loss of M-CSF production and increased cell death ultimately inhibiting development of long-lived ApoE⁺CD11b⁺ AMs upon intranasal β-glucan challenge. ApoE⁺ MoMacs are also found in white adipose tissues during obesity, a condition conferring a training-like feature to MPs or during influenza-induced lung inflammation supporting the crucial role of ApoE for the development of functionally adapted macrophages during inflammation[29,60]. Other work has established functional NF-κB-response elements within the *APOE* gene, and CARD9 directly activates NF-κB thus linking activation of inflammatory NF-κB responses to the induction of inflammatory adaptation in MoMacs.

Collectively, we provide evidence that a single non-pathological environmental stimulation via the Dectin-1–CARD9 axis generates inflammation-experienced MoMacs, which modify subsequent pulmonary acute and chronic inflammation under the control of ApoE, thus molecularly linking macrophage inflammatory amplitude to lung resilience and disease susceptibility.

## Online content

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

H. Theobald [1,24], D. A. Bejarano [1,24], N. Katzmarski[1,24], J. Haub[1,24], J. Schulte-Schrepping[2,3], J. Yu[1], K. Bassler [2], A. L. Ament[4], C. Osei-Sarpong[5], F. Piattini[6], L. Vornholz[7,8], W. T'Jonck[9], A. H. Györfi[10,11], H. Hayer[1], X. Yu[12], S. Sheoran[1], A. Al Jawazneh[13,14], S. Chakarov[15], K. Haendler[16,17], G. D. Brown [18], D. L. Williams[19], L. Bosurgi[13,14], J. H. W. Distler[10,11], F. Ginhoux [15,20,21], J. Ruland [7,8,22,23], M. D. Beyer [5,16], M. Greter [12], C. C. Bain [9], A. I. Vazquez-Armendariz[4], M. Kopf[6], J. L. Schultze [2,3,16] & A. Schlitzer [1]✉

[1]Quantitative Systems Biology, Life & Medical Sciences Institute, University of Bonn, Bonn, Germany. [2]Genomics & Immunoregulation, Life & Medical Sciences Institute, University of Bonn, Bonn, Germany. [3]Systems Medicine, Deutsches Zentrum für Neurodegenerativen Erkrankungen (DZNE), Bonn, Germany. [4]University of Bonn, Transdisciplinary Research Area Life and Health, Organoid Biology, Life & Medical Sciences Institute, Bonn, Germany. [5]Immunogenomics & Neurodegeneration, German Center for Neurodegenerative Diseases, Bonn, Germany. [6]Institute of Molecular Health Science, Department of Biology, ETH Zürich, Zürich, Switzerland. [7]Institute of Clinical Chemistry and Pathobiochemistry, School of Medicine and Health, Technical University of Munich, Munich, Germany. [8]TranslaTUM, Center for Translational Cancer Research, Technical University of Munich, Munich, Germany. [9]Centre for Inflammation Research, Institute for Regeneration and Repair, University of Edinburgh, Edinburgh BioQuarter, Edinburgh, UK. [10]Department of Rheumatology, University Hospital Düsseldorf, Medical Faculty of Heinrich-Heine University, Düsseldorf, Germany. [11]Hiller Research Center, University Hospital Düsseldorf, Medical Faculty of Heinrich-Heine University, Düsseldorf, Germany. [12]Institute of Experimental Immunology, University of Zurich, Zurich, Switzerland. [13]I. Department of Medicine, University Medical Center Hamburg-Eppendorf, Hamburg, Germany. [14]Protozoa Immunology, Bernhard Nocht Institute for Tropical Medicine, Hamburg, Germany. [15]Shanghai Institute of Immunology, Shanghai JiaoTong School of Medicine, Shanghai, China. [16]PRECISE Platform for Single Cell Genomics and Epigenomics at DZNE & University of Bonn and West German Genome Center, Bonn, Germany. [17]Institute of Human Genetics, University Medical Center Schleswig-Holstein, University of Luebeck & Kiel University, Luebeck, Germany. [18]MRC Centre for Medical Mycology, University of Exeter, Exeter, UK. [19]Department of Surgery and Center for Inflammation, Infectious Disease and Immunity, James H. Quillen College of Medicine, East Tennessee State University, Johnson City, TN, USA. [20]Singapore Immunology Network, Agency for Science, Technology and Research, Singapore, Singapore. [21]INSERM U1015, Gustave Roussy Cancer Campus, Villejuif, France. [22]German Center for Infection Research (DZIF), partner site Munich, Munich, Germany. [23]German Cancer Consortium (DKTK), partner site Munich, Munich, Germany. [24]These authors contributed equally: H. Theobald, D. A. Bejarano, N. Katzmarski, J. Haub. ✉e-mail: andreas.schlitzer@uni-bonn.de

## Methods

### Animal studies and mouse models

All mice used in this study were bred in the animal facility of the LIMES Institute, University of Bonn, Germany or Center for Translational Cancer Research, Klinikum rechts der Isar, Technical University of Munich, Germany. Mice were housed in individually ventilated cages under conventional conditions (12-h/12-h light–dark cycle, 22 °C), with ad libitum access to food and water. All experiments were performed using C57BL/6J (WT) mice, which also served as controls for *Ccr2*[−/−], *Dectin1*[−/−] and *CARD9*[−/−] knockout mice. For *Apoe*[fl]*Lyz2*[Cre/+] mice, Cre-negative ApoE[fl]*Lyz2*[+/+] littermates were used as controls. Eight- to twelve-week-old male mice were used for experiments. All experiments were approved by the government of North Rhine-Westphalia (84-02.04.2017.A347, 81-02.04.2020.A454).

### Intranasal stimulation

Mice were anesthetized by intraperitoneal injection of a ketamine–xylazine mixture and intranasally inoculated with endotoxin-free 1× PBS (EMD Millipore) or 200 µg β-glucan from *Candida albicans*. BALF was collected 1, 3, 7, 14 or 21 days after inoculation. For cell analysis, the lung was flushed three times with 1 ml cold 1× PBS with 10 mM EDTA. Afterwards, the fluid was centrifuged for 5 min at 365$g$ at 4 °C, and the supernatant was discarded. For cytokine and chemokine assessment, the lung was flushed three times with the same 1 ml of cold 1× PBS with 10 mM EDTA. Afterwards, the supernatant without cells was frozen in liquid nitrogen until analysis. Supernatants were thawed on ice and centrifuged for 5 min at 10,000 rpm at 4 °C to remove debris. M-CSF (R&D Systems) and ApoE (Abcam) protein levels were measured by ELISA according to manufacturer's protocols.

### Macrophage transfer

C57BL/6J CD45.2 WT donor mice were intranasally stimulated with PBS or β-glucan as described above. At day 5, BAL fluid was harvested, and cells of the same condition were pooled. After centrifugation, the supernatant was discarded, and cells were resuspended in LPS-free 1× PBS. Afterwards, $2 \times 10^5$ donor cells in a volume of 35 µl were intratracheally transferred into CD45.1 recipient mice. BALF was harvested 48 h after transfer and either analyzed by flow cytometry or used for ex vivo restimulation with LPS (see below).

### In vivo CSF-1 receptor blockade

Before in vivo application, the amount of the anti-CSF-1R or the isotype control antibody was calculated per cohort. To reduce the application volume, a Concentrator Plus vacuum centrifuge (Eppendorf) was used in V-AQ mode to evaporate off excess liquid to yield a final product in one-quarter of the starting volume. C57BL/6J WT mice were intranasally stimulated with β-glucan mixed with 500 µg anti-CSF-1R (BioLegend) or 500 µg isotype control (BioLegend). At 12 h and 3 days after the initial treatment, intranasal application of 500 µg anti-CSF-1R or the respective isotype control was repeated before analysis at day 7.

### Pulmonary fibrosis and *L. pneumophila* infection

Mice were intranasally stimulated with β-glucan or PBS 7 d before induction of pulmonary fibrosis or infection by *L. pneumophila*. For fibrosis induction, *Streptomyces verticillus* bleomycin (Sigma-Aldrich, 0.75 mg per kg body weight) was administered by intranasal installation. Body weight and health status were scored on a daily basis. Analysis was performed 3 or 14 days after bleomycin application. For bacterial infections, intratracheal application of *L. pneumophila* ($5 \times 10^6$ colony-forming units per mouse) was performed, and mice were killed 2 days after induction. For bacterial load determination, BALF supernatant was plated in duplicates on CYE-plates and grown for 3–4 days at 37 °C in a non-$CO_2$ incubator.

### BM chimeras

BM chimeras were generated by multiple intraperitoneal busulfan (Sigma-Aldrich) injections or irradiation of recipient mice with 10 Gy. Afterwards, $5 \times 10^6$ (busulfan-treated mice) or $1.5 \times 10^6$ (irradiated mice) freshly isolated BM cells from the donor animals were intravenously injected into the recipients. Peripheral blood chimerism was assessed 28 d after reconstitution by flow cytometry. BM chimeras were used in experiments after 8–12 weeks of reconstitution.

### Flow cytometry and cell sorting

For flow cytometry, cells of the bronchoalveolar space were harvested by flushing the lungs with $3 \times 1$ ml ice-cold 1× PBS containing 10 mM EDTA. After centrifugation with 365$g$ for 5 min at 4 °C, cell pellets were resuspended in antibody mix and stained for 35 min at 4 °C. After washing with FACS buffer (1× PBS, 2 mM EDTA, 0.5% BSA (SERVA)), life/death stain was performed using DRAQ7 (BioLegend, 1:1,000 dilution in FACS buffer) for 5 min at room temperature (RT). Red blood cell lysis was performed only if necessary. For the lung tissue, the more segmented lobe was minced and enzymatically digested for 45 min at 37 °C in HBSS (PAN Biotech) supplemented with 10% FCS (Sigma-Aldrich), 0.2 mg ml[−1] collagenase IV (Sigma-Aldrich) and 0.05 mg ml[−1] DNase I (Sigma-Aldrich). Afterwards, the tissue pieces were homogenized with a 19G syringe and filtered through a 70-µm strainer. Red blood cell lysis was performed once for 5 min at RT before life/death stain and acquisition. For blood analysis, blood was collected in 1× PBS with 10 mM EDTA and stained in antibody mix for 35 min at 4 °C. Red blood cell lysis was performed twice for 5 min at RT before life/death stain and acquisition. For BM cells, femurs and tibias were flushed with 1× PBS and stained with antibody mix for 1 h, and red blood cell lysis and life/death stain were subsequently performed. Cells were washed and resuspended in FACS buffer and recorded using a FACS Symphony A5 (Becton Dickinson). FACS data were analyzed using FlowJo v10.8.1 (Becton Dickinson).

Cell sorting was performed using an ARIA III (Becton Dickinson) instrument. Briefly, cells from the lung lavage were stained with antibodies followed by life/death stain. Cell sorting was performed using a 100-µm nozzle into cooled 1.5-ml reaction tubes containing FACS buffer.

The following monoclonal anti-mouse antibodies anti-CD45R (clone RA3-6B2, BioLegend; 1:400 dilution), anti-CD117 (clone 2B8, BioLegend; 1:200 dilution), anti-CD11b (clone M1/70, BioLegend; 1:200 dilution), anti-CD11c (clone N418, BioLegend; 1:200 dilution), anti-CD11c (clone N418, BioLegend; 1:200 dilution), anti-CD135 (clone A2F10, BD Biosciences; 1:200 dilution), anti-CD192 (clone SA203G11, BioLegend or clone 475301, BD Biosciences; 1:200 dilution), anti-CD19 (clone 6D5, BioLegend; 1:200 dilution), anti-CD150 (clone 475301, BioLegend; 1:200 dilution), anti-CD3 (clone 17A2, BioLegend; 1:200 dilution), anti-CD131 (clone JORO 50, BD Biosciences; 1:200 dilution), anti-CD45 (clone I3/2.3, BioLegend or 30-F11, BioLegend/BD Biosciences; 1:200 dilution), anti-CD45.1 (clone A20, BD Biosciences; 1:200 dilution), anti-CD45.2 (clone 104, BD Biosciences; 1:200 dilution), anti-CD16/32 (clone 2.4G2, BD Horizon or clone 93, BioLegend; 1:100 dilution), anti-CD206 (clone C068C2, BioLegend; 1:200 dilution), anti-CD48 (clone HM48-1, BioLegend; 1:100 dilution), anti-CD90 (clone 53-2.1, BioLegend; 1:200 dilution), anti-CD64, (clone X54-5/7.1, BioLegend; 1:100 dilution), anti-CX3CR1 (clone SA011F11, BioLegend; 1:100 dilution), anti-F4/80 (clone BM8, BioLegend; 1:100 dilution), anti-IL-6 (clone MP5-20F3, BD Biosciences; 1:100 dilution), anti-Ly6C (clone HK1.4, BioLegend; 1:200 dilution), anti-Ly6G (clone 1A8, BioLegend/BD Biosciences; 1:200 dilution), anti-MERTK (clone 2B10C42, BioLegend; 1:200 dilution), anti-MHC2 (clone M5/114.15.2, BioLegend/BD Biosciences; 1:200 dilution), anti-NK-1.1 (clone PK136, BioLegend; 1:200 dilution), anti-Ly-6A/E (clone D7, Thermo Fisher Scientific; 1:200 dilution), anti-Siglec-F (clone E50-2440, BD Biosciences; 1:200 dilution), anti-mouse TCR beta chain (clone H57-597,

BioLegend; 1:400 dilution), anti-TER-119, (clone TER-119, BioLegend; 1:200 dilution), anti-CD34 (clone SA376A4, BioLegend; 1:100 dilution), anti-CD335 (clone 29A1.4, BioLegend; 1:200 dilution), anti-CD115 (clone AFS98, BioLegend or clone T38-320, BD Biosciences; 1:100 dilution), anti-CD24 (clone M1/69, BioLegend/BD Biosciences; 1:100 dilution), anti-Sca-1 (clone D7, BioLegend; 1:100 dilution) and anti-CD43 (clone S7, BD Biosciences; 1:200 dilution) were used for flow cytometry or cell sorting.

## PrimeFlow RNA detection by flow cytometry

BALF samples of three stimulated WT mice were pooled, centrifuged at 365$g$ for 5 min at 4 °C and resuspended in antibody mix for surface staining. Samples were transferred to 1.5 ml microcentrifuge tubes provided by the PrimeFlow RNA Assay Kit (Thermo Fisher Scientific) and stained for 35 min at 4 °C. Afterwards, life/death staining was performed using Zombie NIR Fixable Viability dye (1:1,000 dilution in PBS, BioLegend) for 10 min at RT. In the further steps, samples were handled according to the manufacturer's instructions.

## Ex vivo stimulation and assessment of cytokine production

BALF of stimulated mice was collected, centrifuged at 365$g$ for 5 min at 4 °C and resuspended in 1 ml RPMI 1640 (PAN Biotech) supplemented with 10% FCS (Sigma-Aldrich), 2 mM GlutaMAX (Gibco), 1% MEM non-essential amino acids (Sigma-Aldrich), 1 mM sodium pyruvate (Gibco), 50 U ml$^{-1}$ penicillin–streptomycin (Gibco) and 0.1% β-mercaptoethanol. Cells were counted and seeded with $0.2 \times 10^5$ cells per well. After 2 h resting in 500 µl medium at 37 °C and 5% CO$_2$, medium was exchanged to wash away non-adherent cells. Remaining macrophages were subsequently stimulated with 10 ng ml$^{-1}$ LPS (Sigma-Aldrich) in a final volume of 500 µl. For intracellular cytokine stain, cells were restimulated for 4 h, then 2.5 µg brefeldin A (BioLegend) and 2 nM monensin (BioLegend) were added to each well and incubated for a further 2 h. Cells were harvested in 1× PBS using a cell scraper followed by staining of surface markers by antibodies for 30 min at 4 °C. Cells were washed and stained with Zombie NIR fixable viability dye (1:1,000 dilution in PBS, BioLegend) for 15 min. Afterwards, cells were permeabilized using the Cytofix/Cytoperm kit (Becton Dickinson, adapted from the manufacturer's protocol). In brief, cells were resuspended in 200 µl Cytofix/Cytoperm solution per tube and incubated for 20 min at 4 °C. Cells were washed twice with Perm/Wash and intracellularly stained using 100 µl Perm/Wash containing IL-6 (MP5-20F3; 1:100 dilution) antibody or the corresponding isotype control for 30 min at 4 °C. Cells were washed twice with Perm/Wash and resuspended in 1× PBS before acquisition. For cytokine assessment from the supernatant, cell culture supernatant was harvested 24 h after LPS and snap frozen for further analysis. Supernatants were thawed on ice and centrifuged for 5 min at 10,000 rpm at 4 °C to remove debris. IL-6 (Thermo Fisher) protein levels were measured by ELISA. For multiplex cytokine and chemokine analysis, a customized 18-plex Procartaplex kit (Thermo Fisher) was used according to manufacturer's protocols and run on a Luminex FLEXMAP 3D (Thermo Fisher) device.

## In vitro phagocytosis assay

BALF cells of PBS- or β-glucan-inoculated mice were seeded with $0.2 \times 10^5$ cells per well in a 96-well plate and selected by adherence as before. Medium was exchanged to 100 µl medium or 100 µl medium containing 2.5 µg pHrodo *S. aureus* bioparticles (Sartorius) per well. Phagocytosis was monitored every 10 min for 7 h in total using the microscopy-based approach of the Incucyte instrument (Sartorius). Analysis was performed using the Incucyte basic analyzer software in standard mode with two channels (phase and orange).

## Extracellular flux analysis

BALF of two mice was pooled and $0.5–1 \times 10^5$ cells were plated in a 96-well Seahorse plate (Agilent) in Seahorse XF base medium

(Agilent) supplemented with 5% L-glutamine (Sigma-Aldrich), 10% FCS (Sigma-Aldrich) and 50 U ml$^{-1}$ penicillin–streptomycin (Gibco) for 2 h at 37 °C and 5% CO$_2$. Before acquisition, cells were washed and incubated in FCS and glucose-free Seahorse XF base medium with 5% L-glutamine (Sigma-Aldrich) and 50 U ml$^{-1}$ penicillin–streptomycin (Gibco). During the run, 100 mM glucose (Sigma-Aldrich) solution was injected into port A leading to a final glucose concentration of 10 mM per well. This was followed by injection of 10 µM oligomycin A (Sigma-Aldrich, final concentration 1 µm) solution and 500 mM 2-desoxyglucose (Sigma-Aldrich, final concentration 50 mM). Glycolysis, glycolytic capacity and glycolytic reserve were calculated using the Agilent Wave software. After the Seahorse assay, cell numbers per well were determined for normalization using the CyQUANT NF Cell Quantification Assay (Thermo Fisher) and a TECAN plate reader.

## Lung organoid generation, fibrosis induction by TGF-β and immunofluorescence staining

Organoid cultures were prepared as previously described and cultured at 37 °C with 5% CO$_2$ (ref. 39). In brief, lung single-cell suspensions were prepared from adult wild-type mice and CD31$^+$CD45$^+$CD16/32$^+$ cells were depleted by antibody-coupled magnetic beads. From the CD31$^-$CD45$^-$CD16/32$^-$ negative fraction, EpCAM$^{hi}$CD24$^{lo}$Sca-1$^+$ bronchoalveolar stem cells (BASCs) and EpCAM$^-$Sca-1$^+$ lung resident mesenchymal cells (rMCs) were isolated by FACS. $5 \times 10^3$ BASCs and $1.8 \times 10^4$ rMCs were pooled and mixed with growth factor-reduced Matrigel (Corning; 1:1 ratio) and seeded on 12-mm cell culture inserts in a 24-well plate. α-MEM medium (Thermo Fisher) supplemented with 10% FCS (Thermo Fisher), 50 U ml$^{-1}$ penicillin–streptomycin (Thermo Fisher), 1× insulin–transferrin–selenium (Thermo Fisher) and 2 µg ml$^{-1}$ heparin (Stemcell Technologies) was added to the wells to obtain an air–liquid interface. For the co-culture, AMs were obtained from the BALF of WT mice 7 days after PBS or β-glucan inoculation. Subsequently, $2.5 \times 10^4$ AMs were seeded on top of the Matrigel layer of the day 21 organoid cultures. Twenty-four hours later, the organoid–AM co-cultures were treated with medium containing either PBS or 1.05 ng ml$^{-1}$ TGF-β (Miltenyi Biotech) to induce fibrosis. After 48 h of TGF-β treatment, cultures were fixed in 4% paraformaldehyde (PFA), permeabilized and blocked overnight with 1× PBS containing 0.5% Triton X-100 (Thermo Fisher Scientific) and 5% donkey serum (PAN Biotech; blocking buffer). Primary and secondary antibodies were incubated overnight in blocking buffer. The samples were cleared by glycerol–fructose clearing as recently described[61].

## Histology

Mice were anesthetized followed by transcardial perfusion with 10 ml ice-cold 1× PBS containing 10 mM EDTA using the lung–heart circulation. Lungs were removed and fixed in 4% PFA overnight at 4 °C (for paraffin-embedded tissue) or infiltrated with 1 ml 50% OCT compound (in 1× PBS), removed and fixed for 6 h in 1.3% PFA at 4 °C. For paraffin sections, lungs were dehydrated and paraffin embedded. For frozen sections, after fixation lungs were dehydrated in 10%, 20% and 30% sucrose (in 1× PBS) for 24 h at 4 °C. After dehydration, the left lobe was separated and embedded in OCT. Sections of 5 µm were prepared for immunohistochemistry.

## Immunofluorescence and histology staining

Coverslips containing frozen tissue sections were left drying on Drierite beads for 5 min and subsequently fixed on ice-cold acetone for 10 min. Afterwards, sections were washed twice and permeabilized with 0.2% Triton X-100 for 20 min at RT. Afterwards, sections were washed twice with 1× PBS and photobleached as described before[62]. Following photobleaching, sections were blocked in 3% BSA for 1 h at RT. After blocking, primary antibodies were added and left incubating overnight at 4 °C. Sections were then washed three times with 1× PBS and

secondary antibodies and nuclear staining solution were subsequently added and left to incubate for 1 h at RT. Samples were washed as before and coverslips were mounted using mounting medium. Fluorescently labeled primary antibodies were added after washing the secondary antibodies and left to incubate for 2 h at RT. For immunofluorescence of cultured cells, no acetone fixation and photobleaching were performed.

Histological evaluation of lung fibrosis was performed by Picrosirius red and Masson's trichrome stainings of two consecutive paraffin-embedded 5-μm tissue sections as previously described[63,64].

### Filipin, TUNEL and BODIPY-cholesterol staining
In total, $5–8 \times 10^4$ BALF cells were seeded in complete RPMI medium in a 24-well plate containing sterile glass coverslips. Cells were left adhering for 3 h at 37 °C. For filipin and TUNEL stainings, cells were washed with 1× PBS and subsequently fixed with 4% PFA for 30 min. For filipin staining, after washing away the fixative, cells were incubated in 100 mM glycine for 10 min at RT, and subsequently blocked with 3% BSA supplemented with 50 μg ml$^{-1}$ filipin (Sigma-Aldrich) for 2 h at RT. Cells were washed three times with 1× PBS and immunostained as indicated above, but DRAQ5 (Thermo Fisher Scientific) was used as a counterstain. For TUNEL staining, the manufacturer's instructions were followed and immunofluorescence was performed after TUNEL (Thermo Fisher Scientific). For BODIPY-cholesterol staining, BALF cells were seeded in 8-well chamber slides, left as before and incubated overnight in complete RPMI 1640 medium supplemented with 0.5 μM BODIPY-cholesterol (Biomol). Cells were washed three times with 1× PBS, fixed, permeabilized and immunostained as described above.

### Imaging
Images of the Picrosirius red and Trichrome stainings were acquired using the OLYMPUS Slideview VS200 (Evident Corporation). Sections were analyzed at a magnification of ×20. Images of immunofluorescence of tissue sections and cultured cells were acquired using a Zeiss LSM 880 Airyscan system using a ×60 oil immersion objective (NA) with a z-spacing of 500 nm. Images were acquired using the 405, 488, 561 and 640-nm laser lines. During acquisition, nuclei showing the prototypical shape of neutrophils or eosinophils were excluded.

### Image analysis
To quantify signal intensities from different markers from individual BALF cells, images were analyzed with a customized pipeline in Cell-Profiler. Briefly, Hoechst or DRAQ5 signals were used to segment the cells. A second primary detection step was added to create a mask of all Siglec-F$^+$ objects. This mask was subsequently merged onto the nuclei mask and only overlapping objects were further analyzed. A secondary object detection step was incorporated to distinguish between ApoE$^+$CD11b$^+$ and CD11b$^-$ cells and create a mask of AMs. Analysis of immunofluorescence of tissue sections was performed in QuPath[65]. Nuclear signals were used to identify all objects using a radius of 2 μm. For each channel, an object classifier was created to set the detection threshold based on the mean signal intensity. Subsequently, these classifiers were combined to identify AMs. Mean intensities of individual cells were exported. To measure the area of fibrosis and the Ashcroft score, scoring was performed in four different areas of each slide as previously described[66,67]. Quantification of fibrotic area from total tissue area was performed with ImageJ. The Ashcroft score was quantified using scores ranging from 0 to 8 by two independent investigators, which were blinded to the treatments. To score fibrosis in cultured organoids, images were imported into QuPath, and the area of the organoid was annotated using DAPI as a reference. A pixel classifier to detect SMA-positive areas was trained using the control samples (that is, untreated and TGF-β treated). Around 5–10 exemplary regions

of SMA and background spots were selected. The trained model was applied to analyze all the images. The total SMA$^+$ area of each organoid was quantified. To quantify the percentage of colocalization and overlapping area of BODIPY-cholesterol with cellular organelles, full z-stacks of single ApoE$^+$CD11b$^+$ and CD11b$^-$ AMs were uploaded to Fiji and analyzed using the JACoP plugin, as described before[68]. Thresholds were established using five randomly selected images from each condition and then applied to all the images.

### CODEX multiplexed imaging and analysis
Fresh frozen sections of the left lobe of the lung of 8-week-old *Ms4a3-cre*$^{Rosa26TOMATO}$ mice 7 days after intranasal PBS or β-glucan were prepared and stained following manufacturer's instructions. Briefly, sections were fixed in ice-cold acetone for 10 min. Afterwards, samples were rehydrated and permeabilized for 20 min with 0.2% Triton X-100. Sections were photobleached twice for 1 h as indicated before[62]. After photobleaching, samples were equilibrated for 30 min in staining buffer (Akoya Biosciences), and subsequently stained with a 17-plex CODEX antibody panel overnight at 4 °C. After staining, samples were washed in staining buffer, fixed in ice-cold methanol and washed. A final fixation step with BS3 crosslinker (Sigma-Aldrich) was performed. Specimens were stored in CODEX storage buffer (Akoya Biosciences) at 4 °C for a maximum of 1 week before imaging. BALF cells from WT mice were seeded on CODEX coverslips after harvesting 7 days after intranasal stimulation. Two hours after seeding, cells were fixed with 4% PFA for 20 min, washed and stored in PBS at 4 °C until CODEX staining. Except for the initial drying step, the same staining protocol and CODEX panel as for the lung sections were used.

Antibody detection was performed in a multicycle experiment with the corresponding fluorescently labeled reporters, following the manufacturer's instructions. Images were acquired with a Zeiss Axio Observer widefield microscope (Carl Zeiss AG) using a ×20 air objective (NA 0.85) and a z-spacing of 1.5 μm. The 405-nm, 488-nm, 568-nm and 647-nm fluorescence channels were used. After acquisition, images were exported using the CODEX Instrument Manager (CIM, Akoya Biosciences) and processed with the CODEX Processor v1.7 (Akoya Biosciences). Cells were segmented using DAPI signals and ATPase I membrane staining to define the cell borders. Cell classification to detect AMs and other MPs was performed in CODEX MAV (Akoya Biosciences), following a similar gating scheme to the one used for flow cytometry.

### Preparation of Seq-Well arrays and libraries
Seq-Well arrays and libraries were generated as previously described[69]. Briefly, arrays were generated by pouring PDMS master mix into master molds and then functionalized by plasma treatment, washing with acetone, incubation with 0.2% chitosan solution and subsequent incubation in PGA buffer under vacuum pressure. For library generation, $1.1 \times 10^5$ barcoded mRNA-capture beads in Bead Loading Buffer were loaded onto the array. Around $2–3 \times 10^4$ BALF cells in RPMI 1640 medium (Gibco) with 10% FCS (Sigma-Aldrich) were loaded and rocked for 10 min. The loaded arrays were washed, sealed by polycarbonate membranes under mild vacuum, incubated for 30 min at 37 °C in Agilent clamps (Agilent) and then incubated in a guanidinium-based lysis buffer for 20 min. After incubation in hybridization buffer, the mRNA-capture beads were washed from arrays and collected. Reverse transcription was performed on the bead pellet using a Maxima Reverse Transcriptase reaction (Thermo Fisher) for 30 min at RT followed by 90 min of incubation at 52 °C before stopping the reaction with TE buffer supplemented with 0.01% Tween-20. Excess primers were digested by exonuclease ExoI (New England Biolabs). Beads were counted, and the reverse-transcribed cDNA libraries were amplified in a PCR reaction. After PCR, $2–4 \times 10^4$ beads were pooled and cleaned using AMPure XP beads (Beckman Coulter). The library integrity was assessed using a High Sensitivity D5000 assay (Agilent) for Tapestation 4200 (Agilent).

## Sequencing

The cDNA libraries (1 ng) were tagmented with the prepared single-loaded Tn5 transposase mixed with pre-annealed linker oligonucleotides and afterwards cleaned using MinElute PCR kit (Qiagen) following the manufacturer's instructions. The Illumina indices (Illumina) were added to the tagmented product by PCR and subsequently cleaned by AMPure XP beads (Beckman Coulter). The final library quality was assessed using a High Sensitivity DNA5000 assay (Agilent) and quantified using the Qubit high-sensitivity dsDNA assay (Thermo Fisher). Seq-Well libraries were pooled in equimolar amounts and clustered at a concentration of 1.4 pM with 10% PhiX using High Output v2.1 chemistry (Illumina) on a NextSeq 500 system (Illumina). Paired-end sequencing was performed as follows: custom Drop-Seq Read 1 primer for 21 cycles, 8 cycles for the i7 index and 61 cycles for read 2. Single-cell data were demultiplexed using bcl2fastq2 (v2.20; Illumina). Fastq files were loaded into a snakemake-based data pre-processing pipeline (version 0.31, available at https://github.com/Hoohm/dropSeqPipe)[70].

## scRNA-seq data analysis

Sequencing reads were mapped to the mouse reference genome mm10 using STAR alignment from the Drop-seq pipeline (v2.0.0) as previously described[70]. Next, we assessed the quality of our libraries and excluded cells with low quality (<500 genes per cell), doublets (>3,000 genes per cell) or dead cells (>10% of mitochondrial content). All genes expressed in less than five cells were filtered out.

Cell clustering analysis was performed using the Seurat package (v4.1.1) according to instructions[71]. In brief, the expression data were log normalized with a scale factor of 10,000. After scaling, principal component analysis was performed using the top 2,000 variable genes for a linear dimensional reduction. The first 10 principal components were used to cluster cells by the Louvain algorithm. To obtain an optimal cluster resolution, we set the resolution parameter in the FindClusters function as 0.25 to generate five major clusters, which were visualized after nonlinear dimensional reduction with UMAP. Differentially expressed genes in each cluster were identified by using the default Wilcoxon rank-sum test in the FindAllMarkers function, and were defined with logfc.threshold > 0.25 and min. pct > 0.25.

## Statistics

Statistical analysis and comparison were performed using Prism 10 (GraphPad). Data are shown as the mean ± s.d. Statistical significance was assessed by student's *t*-test (unpaired) or ordinary one-way ANOVA with Tukey's multiple-comparisons test. Survival of animals is displayed in Kaplan–Meier survival curves. A *P* value < 0.05 was considered as statistically significant, exact *P* values are displayed in the figures. Mice were randomly allocated to the control or treatment groups by the investigator. Mouse numbers are indicated as '*n*' in the figure legends, as well as the number of independent experiments.

## Reporting summary

Further information on research design is available in the Nature Portfolio Reporting Summary linked to this article.

## Data availability

The scRNA-seq raw reads and processed data were submitted to the NCBI Gene Expression Omnibus under accession number GSE211575. Source data are provided with this paper.

## Code availability

All code used for data visualization of the scRNA-seq data can be found via GitHub at https://github.com/SchlitzerLab/Trained_immunity_2022 (ref. 72).

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

## Acknowledgements

We thank D. Hüsson and S. Weber (Beyer lab) for technical assistance. This study was funded by the Deutsche Forschungsgemeinschaft (DFG, German Research Foundation) under Germany's Excellence Strategy – EXC2151 – 390873048 (to A.S. and J.L.S.), SFB 1454-P05-432325352 (to A.S.), SFB 1454-P16-432325352 (to M.B.), IRTG 2168 272482170 (to M.B. and J.L.S.), SCHU 950/8-1 (to J.L.S), RU 695/12 -1 (to J.R.), Emmy Noether research grant (SCHL2116/1 to A.S.), the BMBF-funded excellence project Diet–Body–Brain (DietBB; to J.L.S.) and the European Research Council (ERC) under the European Union's Horizon 2020 research and innovation programme (grant agreement no. 834154 (to J.R.); SYSCID 733100 (to J.L.S.)). This research was partly funded by the Wellcome Trust (grant nos. 217163 and 102705), the Medical Research Council Centre for Medical Mycology at the University of Exeter for funding (MR/N006364/2; to G.B.) and a Sir Henry Dale Fellowship jointly awarded by the Wellcome Trust and by the Royal Society (grant no. 206234/Z/17/Z; to C.C.B.). This research was also partly funded by the Swiss National Science Foundation (grant nos. 310030_184915 to M.G. and 310030B_182829 to M.K. and F.P.) and by the National Institutes of Health (NIH; grant nos. GM119197, GM083016 and AI173607 to D.L.W.). For the purpose of open access, the author has applied a CC BY public copyright license to any author accepted manuscript version arising from this submission. W.T. is funded by a UKRI Postdoctoral Guarantee Fellowship (EP/X025071/1).

## Author contributions

Conceptualization: H.T., D.A.B., J.H., N.K. and A.S.; formal analysis and investigation: H.T., D.A.B., J.H., N.K., J.S.-S., J.Y., K.B., A.L.A., C.O.-S., F.P., M.K., L.V., Y.X., M.G., L.B., W.T., C.C.B., S.S., A.A.J., H.H., A.I.V.-A., A.H.G., J.H.W.D., A.A., S.C., K.H., G.B., D.L.W., F.G., J.R., M.D.B. and J.L.S.; writing: H.T., D.A.B., J.H. and A.S.; supervision: A.S.

## Competing interests

The authors declare no competing interests.

## Additional information

**Correspondence and requests for materials** should be addressed to A. Schlitzer.

# Reporting Summary

## Statistics

For all statistical analyses, confirm that the following items are present in the figure legend, table legend, main text, or Methods section.

| n/a | Confirmed | |
|---|---|---|
| ☐ | ☒ | The exact sample size (*n*) for each experimental group/condition, given as a discrete number and unit of measurement |
| ☐ | ☒ | A statement on whether measurements were taken from distinct samples or whether the same sample was measured repeatedly |
| ☐ | ☒ | The statistical test(s) used AND whether they are one- or two-sided<br>*Only common tests should be described solely by name; describe more complex techniques in the Methods section.* |
| ☐ | ☒ | A description of all covariates tested |
| ☐ | ☒ | A description of any assumptions or corrections, such as tests of normality and adjustment for multiple comparisons |
| ☐ | ☒ | A full description of the statistical parameters including central tendency (e.g. means) or other basic estimates (e.g. regression coefficient) AND variation (e.g. standard deviation) or associated estimates of uncertainty (e.g. confidence intervals) |
| ☐ | ☒ | For null hypothesis testing, the test statistic (e.g. *F*, *t*, *r*) with confidence intervals, effect sizes, degrees of freedom and *P* value noted<br>*Give P values as exact values whenever suitable.* |
| ☒ | ☐ | For Bayesian analysis, information on the choice of priors and Markov chain Monte Carlo settings |
| ☒ | ☐ | For hierarchical and complex designs, identification of the appropriate level for tests and full reporting of outcomes |
| ☒ | ☐ | Estimates of effect sizes (e.g. Cohen's *d*, Pearson's *r*), indicating how they were calculated |

*Our web collection on statistics for biologists contains articles on many of the points above.*

## Software and code

Policy information about availability of computer code

| | |
|---|---|
| Data collection | BD Diva, Sartorius Incucyte 2022A, BD Aria III, Tecan Infinite M200, Zeiss Axio Observer, Zeiss LSM 880 Airyscan, Akoya CODEX, Leica Stellaris 8, Illumina NextSeq500, Agilent Wave v2.1, TapeStation Analysis vA.02.01 |
| Data analysis | FlowJo v10.8.1, GraphPad Prism v10, Microsoft Excel v2016, R Studio v1.1.463, R v4.1.3., Tecan i-control v1.12, CODEX MAV, CODEX Processor v1.7.0.6, CellProfiler, QuPath v0.3, bcl2fastq2 v2.20, Drop-seq v2.0.0, Seurat v4.1.1, ImageJ v2 |

For manuscripts utilizing custom algorithms or software that are central to the research but not yet described in published literature, software must be made available to editors and reviewers. We strongly encourage code deposition in a community repository (e.g. GitHub). See the Nature Portfolio guidelines for submitting code & software for further information.

## Data

Policy information about availability of data

All manuscripts must include a data availability statement. This statement should provide the following information, where applicable:
- Accession codes, unique identifiers, or web links for publicly available datasets
- A description of any restrictions on data availability
- For clinical datasets or third party data, please ensure that the statement adheres to our policy

Single-cell RNA sequencing reads were mapped to mouse reference genome mm10. Raw data can be accessed with GEO number "GSE211575". Scripts to generate all results and figures can be found at: https://github.com/schlitzerlab/trained_immunity_2022

# Research involving human participants, their data, or biological material

Policy information about studies with human participants or human data. See also policy information about sex, gender (identity/presentation), and sexual orientation and race, ethnicity and racism.

| Reporting on sex and gender | not applicable |
|---|---|
| Reporting on race, ethnicity, or other socially relevant groupings | not applicable |
| Population characteristics | not applicable |
| Recruitment | not applicable |
| Ethics oversight | not applicable |

Note that full information on the approval of the study protocol must also be provided in the manuscript.

# Field-specific reporting

Please select the one below that is the best fit for your research. If you are not sure, read the appropriate sections before making your selection.

☒ Life sciences ☐ Behavioural & social sciences ☐ Ecological, evolutionary & environmental sciences

For a reference copy of the document with all sections, see nature.com/documents/nr-reporting-summary-flat.pdf

# Life sciences study design

All studies must disclose on these points even when the disclosure is negative.

| Sample size | No statistical method was used to predetermine sample size. Each flow cytometry experiment was performed with at least 3-5 mice per group per replicate, which is commonly accepted in the field of immunology. |
|---|---|
| Data exclusions | Animal experiments: For bone marrow chimeras, mice were excluded from analysis if chimerism was <85%. Flow cytometry: no data was excluded. Single cell RNA sequencing: Cells that did not pass quality control (>10% mitochondrial genes, <500 or >3000 genes per cell) were excluded. |
| Replication | All experiments were successfully replicated and gave similar results. The number of replicates is indicated in the figure legends. |
| Randomization | Mice were randomly assigned to groups by genotype and treatment prior to the initiation of the experiment. |
| Blinding | Sample preparation, data acquisition and analysis was done blinded using the mouse ID. For histological scoring of fibrosis, sections were analyzed by two independent investigators that were blinded to the treatment. Single cell sequencing data analysis was performed by a bioinformatician, who had limited knowledge of the conditions prior to analysis. |

# Reporting for specific materials, systems and methods

We require information from authors about some types of materials, experimental systems and methods used in many studies. Here, indicate whether each material, system or method listed is relevant to your study. If you are not sure if a list item applies to your research, read the appropriate section before selecting a response.

## Materials & experimental systems

| n/a | Involved in the study |
|---|---|
| ☐ | ☒ Antibodies |
| ☒ | ☐ Eukaryotic cell lines |
| ☒ | ☐ Palaeontology and archaeology |
| ☐ | ☒ Animals and other organisms |
| ☒ | ☐ Clinical data |
| ☒ | ☐ Dual use research of concern |
| ☒ | ☐ Plants |

## Methods

| n/a | Involved in the study |
|---|---|
| ☒ | ☐ ChIP-seq |
| ☐ | ☒ Flow cytometry |
| ☒ | ☐ MRI-based neuroimaging |

# Antibodies

Antibodies used

Anti-mouse CD45R , Clone: RA3-6B2, BioLegend, Cat#: 103224, RRID:AB_313007, dilution 1:400
Anti-mouse CD117, Clone: 2B8, BioLegend, Cat#: 105814, RRID:AB_313223, dilution 1:200
Anti-mouse CD11b, Clone: M1/70, BioLegend, Cat#: 101236, RRID:AB_11203704, dilution 1:200
Anti-mouse CD11b, Clone: M1/70, BD Bioscience, Cat#: 612800, RRID:AB_2870127, dilution 1:200
Anti-mouse CD11b, Clone: M1/70, BioLegend, Cat#: 101259, RRID: AB_2566568, dilution 1:200
Anti-mouse CD11b, Clone: M1/70, BioLegend, Cat#: 101243, RRID:AB_2561373, dilution 1:200
Anti-mouse CD11b, Clone: M1/70, BioLegend, Cat#: 101202, RRID:AB_312785, dilution 1:200
Anti-mouse CD11c, Clone: N418, BioLegend, Cat#: 117328, RRID:AB_2129641, dilution 1:200
Anti-mouse CD11c, Clone: N418, BioLegend, Cat#: 117333, RRID:AB_11204262, dilution 1:200
Anti-mouse CD11c, Clone: N418, BioLegend, Cat#: 117338, RRID: AB_2562016, dilution 1:200
Anti-mouse CD11c, Clone: N418, Biolegend, Cat#: 117302, RRID:AB_313771, dilution 1:200
Anti-mouse CD135, Clone: A2F10, BD Bioscience, Cat#: 562537, RRID: AB_2737639, dilution 1:200
Anti-mouse CD192, Clone: SA203G11, BioLegend, Cat#: 150603, RRID: AB_2566139, dilution 1:200
Anti-mouse CD192, Clone: 475301, BD Bioscience, Cat#: 750042, RRID: AB_2874259, dilution 1:200
Anti-mouse CD19, Clone: 6D5, BioLegend, Cat#: 115530, RRID:AB_830707, dilution 1:200
Anti-mouse CD150, Clone: 475301, BioLegend, Cat#: 115910, RRID: AB_493460, dilution 1:200
Anti-mouse CD3, Clone: 17A2, BioLegend, Cat#: 100222, RRID: AB_2242784, dilution 1:200
Anti-mouse CD131, Clone: JORO 50, BD Bioscience, Cat#: 559920, RRID: AB_397374, dilution 1:200
Anti-mouse CD45, Clone: I3/2.3 BioLegend, Cat#: 147710, RRID:AB_2563542, dilution 1:200
Anti-mouse CD45, Clone: 30-F11, BD Bioscience, Cat#: 564279, RRID: AB_2651134, dilution 1:200
Anti-mouse CD45, Clone: 30-F11, BD Bioscience, Cat#: 748370, RRID: AB_2872789, dilution 1:200
Anti-mouse CD45, Clone: 30-F11, BioLegend, Cat#: 103106, RRID: AB_312971, dilution 1:200
Anti-mouse CD45, Clone: 30-F11, eBioscience, Cat#: 14-0451-82, RRID:AB_467251, dilution 1:200
Anti-mouse CD45, Clone: 30-F11, BD Bioscience, Cat#: 553080, RRID:AB_394610, dilution 1:200
Anti-mouse CD45.1, Clone: A20, BD Bioscience, Cat#: 565212, RRID: AB_2722493, dilution 1:200
Anti-mouse CD45.2, Clone: 104, BD Bioscience, Cat#: 564880, RRID: AB_2738998, dilution 1:200
Anti-mouse CD16/32, Clone: 2.4G2, BDHorizon, Cat#: 741229, RRID: AB_2870783, dilution 1:100
Anti-mouse CD206, Clone: C068C2, BioLegend, Cat#: 141723, RRID: AB_2562445, dilution 1:200
Anti-mouse CD48, Clone: HM48-1, BioLegend, Cat#: 103439, RRID: AB_2650824, dilution 1:100
Anti-mouse CD90, Clone: 53-2.1, BioLegend, Cat#: 140325, RRID:AB_2650962, dilution 1:200
Anti-mouse CD64, Clone: X54-5/7.1, BioLegend, Cat#: 139314, RRID: AB_2563904, dilution 1:100
Anti-mouse CD64, Clone: X54-5/7.1, BioLegend, Cat#: 139302, RRID:AB_10613107, dilution 1:100
Anti-mouse CX3CR1, Clone: SA011F11, BioLegend, Cat#: 149035, RRID: AB_2629605, dilution 1:100
Anti-mouse F4/80, Clone: BM8, BioLegend, Cat#: 123141, RRID: AB_2563667, dilution 1:100
Anti-mouse IL-6, Clone: MP5-20F3, BD Biosciences, Cat#: 561367, RRID: AB_10679354, dilution 1:100
Anti-mouse Ly6C, Clone: HK1.4, BioLegend, Cat#: 128036, RRID:AB_2562353, dilution 1:200
Anti-mouse Ly6C, Clone: HK1.4, BioLegend, Cat#: 128037, RRID: AB_2562630, dilution 1:200
Anti-mouse Ly6C, Clone: HK1.4, BioLegend, Cat#: 128021, RRID: AB_10640820, dilution 1:200
Anti-mouse Ly6C, Clone: HK1.4, BioLegend, Cat#: 128016, RRID:AB_1732076, dilution 1:200
Anti-mouse Ly6C, Clone: HK1.4, BioLegend, Cat#: 128005, RRID:AB_1186134, dilution 1:200
Anti-mouse Ly6G, Clone: 1A8, BioLegend, Cat#: 127624, RRID:AB_10640819, dilution 1:200
Anti-mouse Ly6G, Clone: 1A8, BD Bioscience, Cat#: 747072, RRID: AB_2871828, dilution 1:200
Anti-mouse Ly6G, Clone: 1A8, BioLegend, Cat#: 127606, RRID: AB_1236494, dilution 1:200
Anti-mouse Ly6G, Clone: 1A8, BioLegend, Cat#: 127614, RRID: AB_2227348, dilution 1:200
Anti-mouse Ly6G, Clone: 1A8, BD Bioscience, Cat#: 741994, RRID: AB_2871294, dilution 1:200
Anti-mouse MERTK, Clone: 2B10C42, BioLegend, Cat#: 151510, RRID: AB_2832533, dilution 1:200
Anti-mouse MERTK, Clone: 108928, BD Bioscience, Cat#: 747890, RRID: AB_2872352, dilution 1:200
Anti-mouse MHC2, Clone: M5/114.15.2 , BioLegend, Cat#: 107635, RRID:AB_2561397, dilution 1:200
Anti-mouse MHC2, Clone: 2G9, BD Bioscience, Cat#: 750171, RRID: AB_2874376, dilution 1:200
Anti-mouse MHC2, Clone: M5/114.15.2, BioLegend, Cat#: 107622, RRID: AB_493727, dilution 1:200
Anti-mouse MHC2, Clone: M5/114.15.2 , BD Bioscience, Cat#: 750280, RRID: AB_2874471, dilution 1:200
Anti-mouse MHC2, Clone: M5/114.15.2, BioLegend, Cat#: 107602, RRID:AB_313317, dilution 1:200
Anti-mouse NK-1.1, Clone: PK136, BioLegend, Cat#: 108724, RRID:AB_830871, dilution 1:200
Anti-mouse Ly-6A/E, Clone: D7, Thermo Fisher Scientific, Cat#: 45-5981-82, RRID: AB_914372, dilution 1:200
Anti-mouse Siglec F, Clone: E50-2440, BD Biosciences, Cat#: 562757, RRID:AB_2687994, dilution 1:200
Anti-mouse Siglec F, Clone: E50-2440, BD Biosciences, Cat#: 740280, RRID: AB_2740019, dilution 1:200
Anti-mouse Siglec F, Clone: E50-2440, BD Biosciences, Cat#: 565934, RRID: AB_2739398, dilution 1:200
Anti-mouse Siglec F, Clone: E50-2440, BD Biosciences, Cat#: 565527, RRID: AB_2732831, dilution 1:200
Anti-mouse Siglec F, Clone: 1RNM44N, eBioscience, Cat#: 14-1702-82, RRID: AB_2572866, dilution 1:200
Anti-mouse TCR beta chain, Clone: H57-597, BioLegend, Cat#: 109219, RRID: AB_893626, dilution 1:400
Anti-mouse TER-119, Clone: TER-119, BioLegend, Cat#: 116223, RRID:AB_2137788, dilution 1:200
Anti-mouse CD16/32, Clone: 93, BioLegend, Cat#: 101320, RRID:AB_1574975, dilution 1:100
Anti-mouse CD16/32, Clone: 93, BioLegend, Cat#: 101325, RRID:AB_1953273, dilution 1:100
Anti-mouse CD34, Clone: SA376A4, BioLegend, Cat#: 152208, RRID: AB_2650766, dilution 1:100
Anti-mouse CD335, Clone: 29A1.4, BioLegend, Cat#: 137631, RRID: AB_2617040, dilution 1:200
Anti-mouse CD115, Clone: AFS98, BioLegend, Cat#: 135528, RRID: AB_2566523, dilution 1:100
Anti-mouse CD115, Clone: T38-320, BD Bioscience, Cat#: 749974, RRID: AB_2874201, dilution 1:100
Anti-mouse CD115, Clone: AFS98, BioLegend, Cat#: 135523, RRID: AB_2566459, dilution 1:100
Anti-mouse CD24, Clone: M1/69, eBioscience, Cat#: 12-0242-83, RRID: AB_465603, dilution 1:100
Anti-mouse CD24, Clone: M1/69, BD Bioscience, Cat#: 564664, RRID: AB_2716853, dilution 1:100
Anti-mouse CD24, Clone: M1/69, BioLegend, Cat#: 101822, RRID:AB_756048, dilution 1:100
Anti-mouse CD31, Clone: Mec13.3, BioLegend, Cat#: 102514, RRID:AB_2161031, dilution 1:200

Anti-mouse CD326, Clone: G8.8, BioLegend, Cat#: 118218, RRID:AB_2098648, dilution 1:200
Anti-mouse Sca-1, Clone: D7, BioLegend, Cat#: 108120, RRID:AB_493273, dilution 1:100
Anti-mouse CD43, Clone: S7, BD Bioscience, Cat#: 741238, RRID: AB_2870790, dilution 1:200
Anti-mouse CD43, Clone: S7, BD Bioscience, Cat#: 560663, RRID: AB_1727479, dilution 1:200
Anti-mouse NKP46, Clone: 29A1.4, BioLegend, Cat# 137631, RRID:AB_2617040, dilution 1:200
Anti-mouse ApoE, Clone: EPR19392, Abcam, Cat#: ab183597, dilution: 1:50
Anti-mouse GPNMB, Clone: EPR18226-147, Abcam, Cat#: ab234529, dilution: 1:100
Anti-mouse CD326 (Ep-CAM), Clone: G8.8, BioLegend, Cat#: 118202, RRID:AB_1089027, dilution: 1:200
Anti-mouse Alpha Smooth Muscle Actin Antibody, Clone: 1A4, BioLegend, Cat#: 904601 RRID:AB_2565041, dilution: 1:200
Anti-mouse Sodium Potassium ATPase, Clone: EP1845Y, Abcam Cat#: ab76020, RRID:AB_1310695, dilution: 1:100
Anti-mouse CD140a (PDGFRa), Clone: APA5, eBioscience, Cat#: 14-1401-82, RRID:AB_467491, dilution: 1:200
Anti-mouse M-CSF, Clone: Polyclonal, ThermoFisher Scientific, Cat#: PA5-95279, RRID: AB_2807083, dilution: 1:50
Donkey Anti-Rat IgG (H+L) Antibody Cy3, Jackson ImmunoResearch, Cat#: 712-165-153, RRID:AB_2340667, dilution: 1:500
Donkey Anti-Goat IgG Antibody Cy3, Sigma-Aldrich, Cat#: AP180C, dilution: 1:500
Donkey anti-Rabbit IgG (H+L) Antibody Alexa Fluor 488, ThermoFisher Scientific, Cat#: A-21206, RRID: AB_2535792, dilution: 1:500
Donkey anti-Rabbit IgG (H+L) Antibody Alexa Fluor 568, ThermoFisher Scientific, Cat#: A10042, RRID: AB_2534017, dilution: 1:500
Donkey Anti-Rabbit IgG (H+L) Alexa Fluor 647, Abcam, Cat#: ab150075, dilution: 1:500
Donkey anti-Rat IgG (H+L) Antibody Alexa Fluor 488, ThermoFisher Scientific, Cat#: A-21208, RRID:AB_141709, dilution: 1:500
Ultra-LEAF™ Purified anti-mouse CD115 (CSF-1R) Antibody, BioLegend, Cat#: 135541, RRID:AB_2832485
Ultra-LEAF™ Purified Rat IgG2a κ Isotype Ctrl Antibody, BioLegend, Cat#: 400573
Anti-mouse α-SMA, clone: 1A4, Sigma-Aldrich, Cat#: F3777, RRID:AB_476977, dilution: 1:200
Anti-mouse EEA1, Clone: F.43.1, Thermo Fisher Scientific, Cat#: MA514794, RRID:AB_10985824, dilution: 1:100
Anti-mouse LAMP1, Clone: LY1C6, ThermoFisher Scientific, Cat#: MA1164, RRID:AB_2536869, dilution: 1:100
Anti-mouse Calreticulin, Clone: polyclonal, Themo Fisher Scientific, Cat#: PA3900, RRID:AB_325990, dilution: 1:200

| Validation | All antibodies used in this study are commercially available and validated by the manufacturer. Validation data are available on the manufacturer's website. Clones were titrated in house on the tissue of interest and validated using single stain or FMO controls. |
|---|---|

# Animals and other research organisms

Policy information about studies involving animals; ARRIVE guidelines recommended for reporting animal research, and Sex and Gender in Research

| Laboratory animals | All mouse strains used in this study have a C57BL/6 background and were bred in the animal facility of the LIMES Institute, University of Bonn, Germany or Center for Translational Cancer Research, Klinikum rechts der Isar, Technical University of Munich, Germany. Experiments were conducted using male mice aged 8-12 weeks. Mice were housed in IVC mice cages under conventional conditions (12 h/12 h light/dark cycle, 22°C, 55-70% humidity), with ad libitum access to food and water. Wildtype C57BL/6J mice are bred in house. Ai14 (JAX: 007914), CCR2ko (JAX: 004999), CD45.1 (JAX: 002014), Dectin-1ko (JAX: 012337) and LysM Cre (JAX: 004781) animals were obtained from The Jackson Laboratories. ApoE flox mice were kindly provided by Prof. J. Heeren, CARD9ko mice were kindly provided by Prof. J. Ruland, Ms4a3Cre mice were kindly provided by Prof. F. Ginhoux. |
|---|---|
| Wild animals | No wild animals were used in this study. |
| Reporting on sex | Male mice were housed in IVC mice cages under conventional conditions (12 h/12 h light/dark cycle, 22°C), with ad libitum access to food and water. |
| Field-collected samples | No field-collected samples were used in this study. |
| Ethics oversight | All experiments were approved by government of North Rhine-Westphalia (licenses 84-02.04.2017.A347 and 81-02.04.2020.A454). |

Note that full information on the approval of the study protocol must also be provided in the manuscript.

# Plants

| Seed stocks | *Report on the source of all seed stocks or other plant material used. If applicable, state the seed stock centre and catalogue number. If plant specimens were collected from the field, describe the collection location, date and sampling procedures.* |
|---|---|
| Novel plant genotypes | *Describe the methods by which all novel plant genotypes were produced. This includes those generated by transgenic approaches, gene editing, chemical/radiation-based mutagenesis and hybridization. For transgenic lines, describe the transformation method, the number of independent lines analyzed and the generation upon which experiments were performed. For gene-edited lines, describe the editor used, the endogenous sequence targeted for editing, the targeting guide RNA sequence (if applicable) and how the editor was applied.* |
| Authentication | *Describe any authentication procedures for each seed stock used or novel genotype generated. Describe any experiments used to assess the effect of a mutation and, where applicable, how potential secondary effects (e.g. second site T-DNA insertions, mosiacism, off-target gene editing) were examined.* |

# Flow Cytometry

## Plots

Confirm that:

☒ The axis labels state the marker and fluorochrome used (e.g. CD4-FITC).

☒ The axis scales are clearly visible. Include numbers along axes only for bottom left plot of group (a 'group' is an analysis of identical markers).

☒ All plots are contour plots with outliers or pseudocolor plots.

☒ A numerical value for number of cells or percentage (with statistics) is provided.

## Methodology

| | |
|---|---|
| Sample preparation | BALF samples were obtained by flushing the lung 3x with 1 ml 1x PBS with 10 mM EDTA. Peripheral blood was collected into 3 ml 1x PBS supplemented with 10 mM EDTA. For bone marrow, one tibia and femur were flushed with FACS buffer and filtered through a 70 μm strainer. Lungs were removed and one half was digested for 45 min at 37° C in HBSS supplemented with 10% FCS, 0.2 mg/ml collagenase IV and 0.05 mg/ml DNase I and then manually homogenized and filtered through a 70 μm strainer.<br>Pellets of single cell suspensions were resuspended in the antibody mix and incubated at 4°C. For the bone marrow, lung and blood, red blood cell lysis was perfomed. All samples were stained for life/death with DRAQ7 (1:1000 in FACS buffer) before acquisition. |
| Instrument | BD FACS Symphony A5, BD Aria III for sorting |
| Software | BD FACS Symphony: Diva software version 9.1, BD Aria III: Diva software 8.0.1 |
| Cell population abundance | After cell sorting, the purity of the sorted populations was >90% |
| Gating strategy | For all analyzed tissues, cells were selected based on FSC-A and SSC-A. Doublets were excluded by FSC-A vs. FSC-H and SSC-A vs. SSC-H. CD45+ immune cells were selected by SSC-A vs. CD45, followed by exclusion of dead DRAQ7+ and lineage (B220, CD19, CD3ε, Nk1.1,Ter-119, TCR-beta, Nkp46 in APC-Cy7)+ cells. From CD45+ Lin- DRAQ7- cells, the respective populations were gated.<br>In the lung and BALF, AM were identified by CD64, MerTk and SiglecF expression. Gating strategy for further separation into CD11b- and CD11b+ AM is included in the supplementary information. According to this strategy, CD11b- and CD11b+ AM were sorted for the in vitro restimulation assay with LPS.<br>For single cell RNA sequencing, SSC high, Lin- (B220, CD19, CD3ε, Nk1.1, Ter-119), DRAQ7- singlets were sorted from the BALF. |

☒ Tick this box to confirm that a figure exemplifying the gating strategy is provided in the Supplementary Information.

