## [Peer Review File · Nature Immunology]

Peer Review Information

Journal: Nature Immunology

Manuscript Title: Apolipoprotein E controls the development of monocyte-derived alveolar macrophages upon pulmonary inflammatory adaptation

Corresponding author name(s): Professor Andreas Schlitzer

Reviewer Comments & Decisions:

Decision Letter, initial version:
--

17th Jan 2023

Dear Dr. Schlitzer,

We have now finished reviewing your response to reviewers comments for your manuscript entitled "Apolipoprotein E controls the development of monocyte-derived alveolar macrophages upon pulmonary inflammatory adaptation", reference number NI-A34954.

We were somewhat concerned that your response to R#1 point #1 would not satisfy the reviewers request. Therefore, we cannot support the publication of your paper in Nature Immunology with the revisions you describe. If you were to perform further experiments to show that alterations in macrophages are required for in the altered responsiveness to infection or fibrosis we may consider resubmission.

We realize that this is disappointing and I apologize for the unusual delay in our response. Please let me know if you would like to discuss this further.

Sincerely,

Stephanie Houston
Editor
Nature Immunology

Reviewers' comments:

Reviewer #1 (Remarks to the Author):

In this paper by Theobald et al. the authors use a model of intranasal Beta-glucan exposure to attempt to understand how lung macrophages respond to physiologic stimuli (fungal ligands are a common environmental exposure). Understanding how monocyte-derived and resident lung macrophages interplay, particularly in less inflammatory environments is of high research relevance to mucosal immunology. This is a clearly written paper with well presented figures. The authors find that, similar to more inflammatory stimuli, Beta-glucan induces recruitment of monocyte-derived macrophages to the lung that share features with resident alveolar macrophages (e.g. high Siglec-F expression). These macrophages have very different functionality to resident alveolar macrophages and in particular produce higher levels of IL-6 and express ApoE. Of note, Beta-glucan exposure results in altered responses to fibrotic damage induced by bleomycin implying that stimulation by environmental-derived signals can alter the tone of the lung tissue. Furthermore, the authors demonstrate that the functionality of the recruited monocyte-derived macrophages is driven by Dectin-1/Card9 and ApoE and in particular that ApoE deficiency is associated with reduced M-CSF production implicating this as the mechanism of action.

The following concerns should be addressed:

- 1) In Fig. 4 the authors demonstrate altered responses of environmentally adapted lung tissue to secondary infection/damage. This is certainly an interesting finding, however, they do not directly link this to altered responsiveness of macrophages specifically. Is it possible to better demonstrate that the effects are due to alterations in the macrophage compartment? e.g. utilising cell transfer or transgenic animals.
- 2) The data presented in Fig. 6 strongly suggests that the impact of ApoE is largely mediated locally. Are there any alterations systemically to the generation of monocytes in these animals, as shown in Fig 2 for WT mice?
- 3) In Fig. 7 the suggestion is that M-CSF is the major mediator of alterations to macrophage populations in ApoE mice. What happens if you block M-CSF in Beta-glucan treated WT animals – do you phenocopy the ApoE^{flox}LysM^{Cre} mice?
- 4) It is unclear to me what is being used as Control mice for the ApoE^{flox} LysM^{Cre} experiments. Could the authors include more details e.g. littermates, ApoE^{flox} Cre negative etc.
- 5) While fungal-derived factors can be environmental stimuli they can also represent pathogenic stimuli from fungal infection itself. There are studies that have investigated some of the pathologic effects of fungal infection on the lung myeloid compartment. Could the authors discuss a little more clearly their environmental stimuli model in the context of this pathologic work.
- 6) While mouse numbers are included in all Figures it is not clear how many independent experiments they represent. Please include.

Reviewer #2 (Remarks to the Author):

Specific comments for the authors to consider in revision:

1. Results section is missing important specific information, such as how many mice or how many cells. This must be included.
2. It's unclear why the authors include monocytes in their supplemental but not in the main. Given their claim that they are monocyte-derived, wouldn't it make sense to include that in the myeloid subcluster analysis?
3. There are better statistical ways to measure altered cell abundance that they should use (Fig 1E and S1F)
4. Not sure why Fig 1G and S1H are looking at total cells instead of a normalized number (percentage?) assuming they get a different number of cells from the BALF each time. The overall increase of Cd11b (S1H) muddies the water a bit on which populations are increasing.
5. The jump from CODEX images to the conclusion of co-expression of markers is missing something that links the observations. I'm not seeing an easy connection here.
6. The unidentified large population in single cell data concerns me (Fig S1I).
7. The claim is made that CCR2 KO mice have a significant reduction in APOE+Cd11b+ AMs compared to WT (only marked as Cd11b in the figure?) but no statistical comparison is shown in Fig 2E & line 158. In fact, there appears to still be an expansion of the cells in CCR2 KO visually. Statistical analysis is needed here.
8. For in vitro stimulation, they mention enrichment of macrophages by adherence but don't mention any washing in the methods. Did they wash away non-adherent cells? Even still, they would most likely also capture monocytes which does not allow them to conclude that IL6 and TNF are produced by the AM alone. This is maybe confirmed by their addition of CCR2 KO showing no significant increase in IL6. They even cite a paper in the discussion that says monocytes show increased IL6 and TNF and say it's in line with their findings.
9. Hard to believe the bleomycin data given the spread of the fibrotic area % in Fig 4D. No PSR or hydroxyproline staining. The authors conclude that exposure to environmental factors can help prevent lung fibrosis which doesn't make sense. A biochemical measure of fibrosis such as hydroxyproline assay is absolutely needed to support the claim.
10. Fig 5: again, showing total cell numbers instead of %. Especially given the spread and the apparent outliers, not sure why.
11. Fig 6: Disconnect between earlier data showing the Cd11b AMs peaking at day 7 (Fig 1G) but APOE protein highest at days 1 and 3 (Fig 6 B).
12. Why the switch to day 3 in Fig 7 when everything else has been day 7?
13. I question whether Fig 7D is truly representative of the norm. Fig 7E suggests an outlier of a few might be driving the statistics. How many times has this been performed?
14. No mention of GMCSF, just MCSF. Have the authors looked at GMCSF? They mention a paper in the discussion that concludes APOE regulates GMCSF sensitivity but don't test it in their model.

Specific comments on the scRNAseq analysis:

1. Why did the authors choose log normalization over SCTransform?
2. Code on github doesn't match methods with regard to PCA dimensions used.
3. Still missing how they went from all cells to just macrophages and why/how the rest of the myeloid cells were left out.

Finally, I'm curious as to whether they believe the BALF reflects all that's occurring in the tissue?

Author Rebuttal to Initial comments

See inserted PDF

We thank the reviewers for their detailed feedback and would like to provide a detailed point-by-point response:

Reviewer #1

The following concerns should be addressed:

1) In Fig. 4 the authors demonstrate altered responses of environmentally adapted lung tissue to secondary infection/damage. This is certainly an interesting finding, however, they do not directly link this to altered responsiveness of macrophages specifically. Is it possible to better demonstrate that the effects are due to alterations in the macrophage compartment? e.g. utilising cell transfer or transgenic animals.

Answer: To address the question raised by the reviewer we have conducted three novel experiments, which have been added to the manuscript.

1. We have flow cytometrically purified either CD11b⁻ or ApoE⁺CD11b⁺ alveolar macrophages from the broncho-alveolar lavage of either PBS or β -glucan stimulated mice. Subsequently these cellular subsets have then been restimulated *in vitro* with LPS. These data show that only ApoE⁺CD11b⁺ AMs from mice which have received a β -glucan stimulus 7 days before exhibit an increased release of IL-6 after LPS restimulation (**Figure 3B, revised manuscript**). No other subsets showed an increased ability to release IL-6 upon LPS restimulation.
2. Our data clearly shows, utilizing both CCR2 deficient and Ms4a3^{Cre} fate mapper mice, that ApoE⁺CD11b⁺ AMs are monocyte-derived. To delineate whether the observed IL-6 production post LPS restimulation is dependent on monocyte recruitment and development of Ly6c⁺ monocytes into ApoE⁺CD11b⁺ AMs, we purified BAL-resident macrophages from either WT or CCR2 deficient mice and restimulated those AMs *ex vivo* with LPS for 24h. Here only mice with intact CCR2 signalling were able to mount enhanced IL-6 responses upon *ex vivo* LPS restimulation (**Figure 3D, revised manuscript**).
3. In order to further establish a direct link of ApoE⁺CD11b⁺ AMs to increased IL-6 production in mice stimulated with β -glucan seven days prior, we intratracheally transferred AMs from mice, which have been stimulated with β -glucan 5 days prior, into naïve mice. We subsequently isolated AMs from those recipients 48hs post transfer and subsequently restimulated them *ex vivo* with LPS for 24h. Here, only in supernatants derived from animals, which had received a cell transplant from mice stimulated with β -glucan seven days prior, had increased level of IL-6 in their restimulated supernatants (**Figure 3E, revised manuscript**).

Thus these three lines of evidence clearly show that CCR2-dependent monocyte-derived ApoE⁺CD11b⁺ AMs present seven days post intranasal β -glucan stimulation exhibit an enhanced ability to secrete increased amounts of IL-6 as compared to other resident AM subpopulations.

2) The data presented in Fig. 6 strongly suggests that the impact of ApoE is largely mediated locally. Are there any alterations systemically to the generation of monocytes in these animals, as shown in Fig 2 for WT mice?

Answer: We thank the reviewer for this important point. We have added new data to the manuscript in order to clarify the role of ApoE in monocyte development. To do so we intranasally stimulated ApoE^{fllox}LysM^{Cre/+} mice or their respective Cre negative littermates with β -glucan and flow cytometrically analysed the blood and bone marrow of these animals seven days post treatment. Ly6c⁺ monocytes of control (ApoE^{fllox}LysM^{+/+}) animals showed decreased cell numbers in the blood (**Figure S6C, revised manuscript**) and increased cell numbers in the bone marrow (**Figure S6D, revised manuscript**) confirming the results we obtained earlier in C57BL/6J mice (**Figure 2E, F, revised manuscript**). These cell number changes were not present in ApoE^{fllox}LysM^{Cre/+} mice (**Figure S6C, D, revised manuscript**). No differences in cMOPs and GMPs were observed in the bone marrow (**Figure S6E, F, revised manuscript**). Importantly numbers of BALF-resident Ly6c⁺ monocytes were unchanged between control and ApoE^{fllox}LysM^{Cre/+}, whereas differentiation into ApoE⁺CD11b⁺ AMs was defective, indicating a local action of ApoE within the broncho-alveolar compartment (**Figure S6A, B, revised manuscript**).

3) In Fig. 7 the suggestion is that M-CSF is the major mediator of alterations to macrophage populations in ApoE mice. What happens if you block M-CSF in Beta-glucan treated WT animals – do you phenocopy the ApoE^{flox}LysM^{Cre} mice?

Answer: We thank the reviewer for this question. To address this point, we blocked M-CSF receptor (CSF-1R) signalling using an *in vivo* anti-CSF-1R monoclonal antibody upon intranasal β -glucan stimulation in C57BL/6J mice (**Figure S7F, revised manuscript**). In presence of the anti-CSF-1R antibody, ApoE⁺CD11b⁺ AM numbers are significantly reduced compared to the group treated with the respective isotype control seven days after β -glucan inoculation (**Figure S7J, revised manuscript**). Other mononuclear phagocyte subsets like CD11b⁻ AMs and Ly6c⁺ monocytes were not affected by the CSF-1R signalling blockade (**Figure S7H, I, revised manuscript**). These results confirm that M-CSF is one of the crucial mediators of the survival/generation of ApoE⁺CD11b⁺ AMs upon intranasal β -glucan stimulation.

4) It is unclear to me what is being used as Control mice for the ApoE^{flox} LysM^{Cre} experiments. Could the authors include more details e.g. littermates, ApoE^{flox} Cre negative etc.

Answer: We thank the reviewer for these important comments and apologize for not being clear on this issue. In the revised manuscript, we added this information to the methods part (**Line: 437f**). In general, all experiments were performed using wildtype C57BL/6J mice, which also served as controls for CCR2, Dectin-1 and CARD9 knockout mice. In case of the ApoE^{flox}LysM^{Cre/+} mice, Cre-negative ApoE^{flox}LysM^{+/+} littermates were used as controls.

5) While fungal-derived factors can be environmental stimuli they can also represent pathogenic stimuli from fungal infection itself. There are studies that have investigated some of the pathologic effects of fungal infection on the lung myeloid compartment. Could the authors discuss a little more clearly their environmental stimuli model in the context of this pathologic work.

Answer: Very good point! We have added these papers in the discussion of the revised manuscript (**Line: 375-378**).

6) While mouse numbers are included in all Figures it is not clear how many independent experiments they represent. Please include.

Answer: Thank you for raising this point and we apologize for this oversight. The figure legends of the revised manuscript now contain the mouse numbers used per group, the number of independent experiments and the statistical test used to assess statistical significance (**Line 748 onwards**).

Reviewer #2

Specific comments for the authors to consider in revision:

1. Results section is missing important specific information, such as how many mice or how many cells. This must be included.

Answer: We thank the reviewer for this comment and apologize for this oversight. In the figure legends of the revised manuscript, we included the mouse numbers per group. For the single-cell data presented in Figure 1, three mice per group were pooled and cells were enriched for SSC^{hi}, Lin⁻ (B220, CD19, CD3 ϵ , Nk1.1, Ter-119), DRAQ7⁻ singlets. 10.505 cells were detected pre-quality control of which 10.202 cells passed quality control parameters and were selected for the final analysis object. We included this information in the figure legend of the revised manuscript (**Line 751-757**).

2. It's unclear why the authors include monocytes in their supplemental but not in the main. Given their claim that they are monocyte-derived, wouldn't it make sense to include that in the myeloid subcluster analysis?

Answer: The presented single-cell transcriptomic analysis of the bronchoalveolar resident macrophages in Figure 1 is not a subcluster analysis and no selection except the exclusion of low-quality cells was done during data processing as outlined above. To generate the single-cell transcriptomic data, bronchoalveolar resident cells from day 7 PBS or β -glucan stimulated mice were flow cytometrically enriched for SSC^{hi} Lin⁻ (B220, CD19, CD3 ϵ , Nk1.1, Ter-119), DRAQ7⁻ singlets prior to processing. Thus, monocytes, which are SSC^{low} are excluded from our single-cell transcriptomic data. We believe that the supplemental data the reviewer is referring to is the UMAP in Figure S1I (**Figure S1O, revised manuscript, Line 900**), which is derived from flow cytometric analysis of the whole lung and not single-cell transcriptomic data. We apologize for not stating this clearly and added more details to the manuscript. We thank the reviewer for pointing this out.

3. *There are better statistical ways to measure altered cell abundance that they should use (Fig 1E and S1F)*

Answer: We thank the reviewer for this comment. The quantitative nature of single cell transcriptomic datasets is debatable. Thus within the manuscript we utilize the dataset in a more qualitative fashion in order to generate hypothesis and drive biological validation of the observed quantitative effects. We believe the relative quantification used for the single cell transcriptomics data in Figure 1E is sufficient as we now have included a direct linkage of ApoE expression on CD11b⁺ AMs, clearly showing that these cells correspond to single cell transcriptomic cluster 3 and provide absolute and relative quantification of all flow cytometry analysis across the complete manuscript validating the initial single cell transcriptomics quantification (**Figure 1E, F-I, revised manuscript**)

4. *Not sure why Fig 1G and S1H are looking at total cells instead of a normalized number (percentage?) assuming they get a different number of cells from the BALF each time. The overall increase of Cd11b (S1H) muddies the water a bit on which populations are increasing.*

Answer: Indeed, the data presented in Figure 1G (**Figure 1H, revised manuscript**) and S1H (**Figure S1J, revised manuscript**) are absolute cell numbers. We agree that a normalized display has excellent value and included percent (%) data for all analysed cell populations across the manuscript confirming our absolute cell number data (e.g. **Figure 1I, S1K, S2A, S6A, revised manuscript**).

5. *The jump from CODEX images to the conclusion of co-expression of markers is missing something that links the observations. I'm not seeing an easy connection here.*

Answer: We agree with the observation of the reviewer and have added 3 additional datasets to Figure 1 in order to make this connection easier to understand.

Firstly, we have set out to directly link ApoE mRNA expression with CD11b protein expression. To do so we used prime flow analysis of BALF resident cells. This analysis shows that the only cells which express ApoE mRNA are CD11b⁺ AMs in mice stimulated with β -glucan seven days prior (**Figure 1G, revised manuscript, Line: 140, 762**). To further allow easier delineation of co-expression of CD11b and ApoE on the protein level we added higher magnification panels of a lung co-detection by indexing (CODEX) (**Figure 1F, revised manuscript, Line: 135, 758**). Filled triangles in Figure 1F highlight cells co-expressing ApoE and CD11b at the protein level, confirming our prime flow data (**Figure 1G, revised manuscript, Line: 138, 758**). Finally, we visualize ApoE and CD11b co-expression on BALF resident cells, showing that cells tagged as CD11b⁺ co-express ApoE (**Figure S1G, revised manuscript, Line: 138, 886**). Taken together we believe that this additional data convinces the reviewer that ApoE in conjunction with CD11b provides comprehensive means to further delineate β -glucan generated monocyte-derived AMs throughout the remainder of the manuscript.

6. *The unidentified large population in single cell data concerns me (Fig S1I).*

Answer: We thank the reviewer for this comment, however, S1I (**Figure S1O, revised manuscript, Line: 900**) is not single-cell transcriptomics data but flow cytometry data. As flow cytometry data cannot account for the complexity of the lung, our myeloid cell-focused staining mix cannot accurately identify these, most likely lymphoid cells. To further show that these unidentified cells are most likely lymphoid cells that were not tagged by our lineage antibodies (B220, CD19, CD3 ϵ , Nk1.1, Ter-119, NKP46, TCR- β) we selected the cluster of unidentified cells from the UMAP and checked their FSC, SSC and marker expression (**Figure A**). The unidentified cells are small with a

low granularity and they are negative for classical myeloid cell markers (MerTK, CD11c, Ly6c, Ly6g, F4/80, SiglecF). They express CD45, CD43 and also partially CD11b and MHC-II. Compared to AMs, they express no CD64 and are not stained by our lineage antibody mix. Further dynamics in this population seems minimal upon β -glucan exposure, thus we think that further delineation is beyond the scope of this work.

Figure A: Unidentified cells in the flow cytometry-based UMAP are negative for myeloid markers and most likely of lymphoid origin. Unidentified clusters were gated and their SSC, FCS and marker expression was investigated.

We thank the reviewer for pointing out this ambiguity. We clearly stated in the figure legends and the results section that this data is based on flow cytometry.

7. The claim is made that *CCR2* KO mice have a significant reduction in *APOE*+*Cd11b*+ AMs compared to WT (only marked as *Cd11b* in the figure?) but no statistical comparison is shown in Fig 2E & line 158. In fact, there appears to still be an expansion of the cells in *CCR2* KO visually. Statistical analysis is needed here.

Answer: As suggested by the reviewer we added more biological replicates and further statistical analysis to this line of experiments (**Figure 2G, revised manuscript, Line: 181, 780**). These data show that a strong reduction in the generation of *ApoE*+*CD11b*+ AM is observed in the absence of intact *CCR2* signalling. Thus confirming our initial observation that *ApoE*+*CD11b*+ AM are monocyte-derived (**Figure 2B-D, revised manuscript, Line: 169, 773 onwards**).

8. For *in vitro* stimulation, they mention enrichment of macrophages by adherence but don't mention any washing in the methods. Did they wash away non-adherent cells? Even still, they would most likely also capture monocytes which does not allow them to conclude that *IL6* and *TNF* are produced by the AM alone. This is maybe confirmed by their addition of *CCR2* KO showing no significant increase in *IL6*. They even cite a paper in the discussion that says monocytes show increased *IL6* and *TNF* and say it's in line with their findings.

Answer: It is correct that restimulation experiments were conducted with AMs selected using adherence. We have now described more in detail the adherence protocol used in this study in the material section, we apologize for this oversight. Further to clarify the cell proportions within our cell adherence selected AM preparations we conducted a flow cytometry to quantify the relative contributions of cell types found within adherence selected AM preparations. Here we found that after adherence selection 89.9±3.3% were AMs, 3.0±0.3% were monocytes, 1.8±1.1% Neutrophils. These data clearly show that a) selection of AMs via adherence is an effective way to enrich for AMs and b) contamination of monocytes is minimal in all adherence selected cell preparations and thus to the released detected cytokines upon restimulation.

B Day 7, BALF after adherence selection

Figure B: Flow cytometric profiling of the cellular composition after selection adherence of macrophages from the BALF after PBS or β -glucan. 0.2×10^5 BALF cells were seeded and then rested for 2h. Afterwards, a washing step was performed to wash away non-adherent cells (n=4-6 mice, one representative experiment).

9. Hard to believe the bleomycin data given the spread of the fibrotic area % in Fig 4D. No PSR or hydroxyproline staining. The authors conclude that exposure to environmental factors can help prevent lung fibrosis which doesn't make sense. A biochemical measure of fibrosis such as hydroxyproline assay is absolutely needed to support the claim.

Answer: We thank the reviewer for raising this important point regarding the fibrosis data. We have now included a more thorough quantification of the fibrotic area using Masson's Trichrome and PSR stains quantifying the % of the fibrotic area and the Ashcroft score. Both analysis show no difference between PBS and β -glucan pre-treated mice in regards to both parameters mentioned. However our assessment of disease burden, survival and weight gain clearly points to an improved outcome in mice pre-treated with β -glucan (**Figure S4G-I, revised manuscript, Line: 236, 914 onwards**). Thus to further understand the role of ApoE⁺CD11b⁺ AM generated in β -glucan pre-treated mice further we generated broncho-alveolar lung organoids and co-cultured either AMs from PBS or β -glucan stimulated mice (seven days post treatment) for 48h with day 21 mature organoids and subsequently stimulated those organoids with TGF- β for 24h to induce a fibrotic reaction. These data show that organoids co-cultured with AMs isolated from mice pre-stimulated with β -glucan seven days prior showed less production of smooth muscle actin (SMA) post TGF- β treatment, whereas organoids co-cultured with AMs from mice pre-stimulated with PBS seven days prior behaved similar to non-co-cultured lung organoids (**Figure 4G, H, S4O, revised manuscript, Line: 245, 815 onwards**). At the level of organoids these co-cultures show that cell-intrinsic properties of AMs isolated from seven days prior β -glucan stimulated mice are beneficial in regards to SMA production upon a fibrotic challenge. Although not directly, we believe that these data strengthen our conclusion that ApoE⁺CD11b⁺ AM improve the outcome of bleomycin induced lung fibrosis, most likely through means not directly affecting the expansion/repair of the fibrotic area within the bleomycin *in vivo* model but likely by modulating inflammation during the early course of disease. To more accurately reflect this data we have amended the text in the revised manuscript (**Figure 4G, H, S4G-N, revised manuscript, Line: 236, 941 onwards**)

10. Fig 5: again, showing total cell numbers instead of %. Especially given the spread and the apparent outliers, not sure why.

Answer: We have now provided all data in a normalized (%) and absolute format (**Figure S5A, C, revised manuscript, Line: 268, 826 onwards**).

11. Fig 6: Disconnect between earlier data showing the Cd11b AMs peaking at day 7 (Fig 1G) but APOE protein highest at days 1 and 3 (Fig 6 B).

Answer: Monocyte-derived macrophages develop from monocytes and the differentiation of monocytes to macrophages in the lung takes seven days within the context of the lung¹. Thus, we do not think that the abundance of ApoE on days 1 and 3 is a disconnection, but rather implies that ApoE is a critical regulator of monocyte to ApoE⁺CD11b⁺ AM development. This is further evidenced by the fact that myeloid-specific ApoE knockout mice stimulated with β -glucan analysed on day 3 show no differences to control mice (**Figure 7A, revised manuscript**). Myeloid-specific ApoE knockout mice only start to show differences after day 3 post-stimulation coinciding with the second spike in ApoE production in WT mice. Thus, the kinetics of ApoE reflects developmental steps within the

monocyte to macrophage trajectory, which, in the absence of ApoE terminates after day 3, due to accumulation of cholesterol and lack of M-CSF release leading to enhanced cell death.

12. Why the switch to day 3 in Fig 7 when everything else has been day 7?

Answer: Our previous data demonstrates that ApoE⁺CD11b⁺ AM develop from monocytes and that development peaks on day seven post β -glucan stimulation (**Figure 1H, revised manuscript**). Thus in an effort to understand whether loss ApoE⁺CD11b⁺ AM is due to a developmental block, cell recruitment and/or cellular maintenance we analysed an earlier time point, prior to the peak of ApoE⁺CD11b⁺ AM development. These data showed that on day three after β -glucan stimulation control and myeloid-cell specific ApoE knockout mice showed no detectable differences within the BALF, indicating that the window of action for ApoE is between day three and seven post β -glucan stimulation (**Figure 7 A, B, revised manuscript, Line: 315, 854 onwards**). Thus, we chose to investigate day three post β -glucan exposure to further explore the mechanisms by which ApoE fosters development of monocytes to macrophages. Subsequently this analysis showed that in the absence of ApoE cholesterol accumulates at the endoplasmic reticulum, coinciding with a loss of M-CSF release and the increase of TUNEL⁺ ApoE⁺CD11b⁺ AMs on day three (**Figure 7E-I, revised manuscript, Line: 327, 860 onwards**).

13. I question whether Fig 7D is truly representative of the norm. Fig 7E suggests an outlier of a few might be driving the statistics. How many times has this been performed?

Answer: The staining provided is the pooled data from two independent experiments with 3 mice in each condition and experiment. This puts the data in a solid foundation to assess cholesterol accumulation in cells. Further subsequent experiments aiming at understating cholesterol trafficking in control versus myeloid-specific ApoE deficient macrophages reveal further differences in regards to intracellular accumulation of cholesterol, thus adding an additional line of the perturbation of cholesterol trafficking and metabolism in ApoE deficient macrophages (**Figure 7E, F, revised manuscript**).

14. No mention of GM-CSF, just M-CSF. Have the authors looked at GM-CSF? They mention a paper in the discussion that concludes APOE regulates GM-CSF sensitivity but don't test it in their model.

Answer: Excellent suggestion. We have looked at the protein level of GM-CSF within the BALF (**Figure C**) and could not detect any GM-CSF increase or dynamics upon β -glucan stimulation, thus concluded that GM-CSF is not playing a crucial role in our model.

Figure C: GM-CSF protein levels in the BALF supernatant at different time points after intranasal β -glucan exposure in wildtype mice (n=4-5 mice, one independent experiment).

Specific comments on the scRNAseq analysis:

1. Why did the authors choose log normalization over SCTransform?

Answer: Both approaches are widely used in single cell mRNA sequencing data analysis. SCTransform is very useful to correct known biases, such as read depth or mitochondrial percentage as indicated in the original SCTransform paper². We do not observe these biases in our dataset (Figure D (A)). In addition, the SCTransform paper has indicated that "both approaches lead to similar results with respect to the major and minor cell populations". Therefore, we do not expect major changes for cell types we determined in present study. To fully address this point, we have performed the SCTransform on our dataset (Figure D (B)). We observe a high degree of correlation of cell populations determined from log normalization and SCTransform (Figure D (C)). In addition, the ApoE⁺CD11b⁺ AMs (cluster 3, SCTransform analysis) are also specifically detected on in the β -glucan stimulated condition, in line with the log normalization results (Figure D (D)). Therefore, the choice of log normalization does not affect our results demonstrated in the manuscript.

Figure D: (A) UMAP colored by sample condition (status), sequencing depth (nFeature_RNA) and mitochondrial percentage (percent.mt). (B) UMAP embedding of cells processed by SCTransform. (C) Comparison of clusters computed from log normalization and SCTransform. Frequency (row-wise) indicates the percentage of log normalization derived clusters in each SCTransform derived clusters. (D) Cell composition in two treatment conditions from the SCTransform analysis.

2. Code on github doesn't match methods with regard to PCA dimensions used.

Answer: We thank the reviewer for pointing out this mistake. We indeed used the code provided in the GitHub repository. For the presented analysis 10 instead of 15 PCA components were used. This typo in the manuscript will be corrected in the revised version of the manuscript. The correct methods part of the single cell transcriptomic data analysis should read: 'After scaling, PCA was performed using the top 2000 variable genes for a linear dimensional reduction. The first 10 PCA components were used to cluster cells by the Louvain algorithm.'

3. Still missing how they went from all cells to just macrophages and why/how the rest of the myeloid cells were left out.

Answer: See answers to questions 1 and 2, Reviewer 1.

Finally, I'm curious as to whether they believe the BALF reflects all that's occurring in the tissue?

Answer: As indicated in the flow cytometry analysis of whole lung digests seven days post PBS or β -glucan challenge (**Figure S10, revised manuscript, Line: 154, 900**), the major differences can be attributed to the formation of monocyte-derived alveolar macrophages. This can be seen by the presence of ApoE⁺CD11b⁺ AM (magenta) population in β -glucan-exposed mice.

References

1. van de Laar, L., Saelens, W., De Prijck, S., Martens, L., Scott, C.L., Van Isterdael, G., Hoffmann, E., Beyaert, R., Saeys, Y., Lambrecht, B.N., and Guilliams, M. (2016). Yolk Sac Macrophages, Fetal Liver, and Adult Monocytes Can Colonize an Empty Niche and Develop into Functional Tissue-Resident Macrophages. *44*, 755-768. [10.1016/j.immuni.2016.02.017](https://doi.org/10.1016/j.immuni.2016.02.017).
2. Hafemeister, C., and Satija, R. (2019). Normalization and variance stabilization of single-cell RNA-seq data using regularized negative binomial regression. *Genome Biol* *20*, 296. [10.1186/s13059-019-1874-1](https://doi.org/10.1186/s13059-019-1874-1).

Decision Letter, first revision:

22nd Jan 2024

Dear Dr. Schlitzer,

Thank you for submitting your revised manuscript "Apolipoprotein E controls the development of monocyte-derived alveolar macrophages upon pulmonary inflammatory adaptation" (NI-A34954A-Z). It has now been seen by the original referees and their comments are below. The reviewers find that the paper has improved in revision, and therefore we'll be happy in principle to publish it in Nature Immunology, pending minor revisions to satisfy the referees' final requests and to comply with our editorial and formatting guidelines.

We will now perform detailed checks on your paper and will send you a checklist detailing our editorial and formatting requirements in about a week. Please do not upload the final materials and make any revisions until you receive this additional information from us.

If you had not uploaded a Word file for the current version of the manuscript, we will need one before beginning the editing process; please email that to immunology@us.nature.com at your earliest convenience.

Thank you again for your interest in Nature Immunology. Please do not hesitate to contact me if you have any questions.

Sincerely,

Stephanie Houston, PhD
Senior Editor
Nature Immunology

Reviewer #1 (Remarks to the Author):

The authors have satisfactorily addressed my concerns.

Reviewer #2 (Remarks to the Author):

In this paper, the authors claim that low grade inflammation in the lung leads to the development of ApoE and M-CSF dependent monocyte derived macrophages that produce IL-6 and are highly phagocytic. The role of ApoE in the development of this population in vitro was confirmed through myeloid cell specific deletion of ApoE. Ultimately, the authors attempted to reveal the relevance of this unique ApoE population in vivo by examining the impacts of b-glucan induced ApoE+ macrophages. They showed that treatment with b-glycan limited L. pneumophila infection and reduced the development of TGFb-driven fibrosis in lung organoids. Overall, while these data are interesting, the system used feels a bit too contrived to definitely verify the importance of ApoE+ AMs. It would have been much more interesting and convincing if the authors showed whether mice deficient in ApoE AM's

are more susceptible to *L. pneumophila* infection and display increased susceptibility to radiation or bleomycin induced fibrosis.

Reviewer #3 (Remarks to the Author):

The current manuscript by Theobald et al identifies a novel functional population of monocyte-derived macrophage population in the lung. The developed a model using low-dose B-glucan administered intranasally to mimic environmental exposures that are thought to establish the immune cell milieu in humans, allowing for the detailed interrogation of molecular mechanisms of pulmonary macrophage adaptation. The importance is that the mechanisms that lead to lung-associated macrophages' functional and developmental adaptation remain elusive. Through a series of single-cell transcriptomics, high dimensional imaging, and flow cytometric characterization paired with in vivo and ex vivo challenge models, the authors identified a unique population of monocyte-derived alveolar macrophages that are Dectin-1 - Card9 signaling-dependent, that expressed high levels of CD11b, ApoE, were glycolytic, and produced large amounts of interleukin 6 upon restimulation. They showed that paracrine derived ApoE is critical for establishment of this population, dependent on M-CSF secretion. Functionally, the authors were able to demonstrate in vivo, that these β -glucan-elicited "MoAMs" limited the bacterial burden of *Legionella pneumophila* post-infection and ameliorated fibrosis severity in a murine fibrosis model. Collectively these data identify MoAMs generated upon environmental cues and ApoE as an essential determinant for lung immune resilience. Together, their conclusions revealed that ApoE controls Dectin-1 -dependent development of monocyte-derived alveolar macrophages upon pulmonary b-glucan-induced inflammatory adaptation.

This paper is well written, well-argued, rigorous, novel, and poised to represent an important advance in the field.

Author Rebuttal, first revision:

See inserted PDF

We thank the reviewers for their feedback and would like to provide a detailed point-by-point response:

Reviewer #1

Remarks to the Author:

The authors have satisfactorily addressed my concerns.

Answer: We thank the reviewer for the positive, constructive feedback and work throughout the publication process.

Reviewer #2

Remarks to the Author:

In this paper, the authors claim that low grade inflammation in the lung leads to the development of ApoE and M-CSF dependent monocyte derived macrophages that produce IL-6 and are highly phagocytic. The role of ApoE in the development of this population in vitro was confirmed through myeloid cell specific deletion of ApoE. Ultimately, the authors attempted to reveal the relevance of this unique ApoE population in vivo by examining the impacts of b-glucan induced ApoE+ macrophages. They showed that treatment with b-glycan limited L. pneumophila infection and reduced the development of TGFb-driven fibrosis in lung organoids. Overall, while these data are interesting, the system used feels a bit too contrived to definitely verify the importance of ApoE+ AMs. It would have been much more interesting and convincing if the authors showed whether mice deficient in ApoE AM's are more susceptible to L. pneumophila infection and display increased susceptibility to radiation or bleomycin induced fibrosis.

Answer: We thank the reviewer for his overall appreciation of the study and his constructive feedback. Concerning the role of ApoE+ AMs during L. Pneumophila, we believe that we have produced sufficient data to prove their important role in protecting against bacterial infection. Thus, we believe that proving the role of ApoE+ AMs directly in our model of ApoE deficiency is beyond the scope of our study.

Reviewer #3

Remarks to the Author:

The current manuscript by Theobald et al identifies a novel functional population of monocyte-derived macrophage population in the lung. The developed a model using low-dose B-glucan administered intranasally to mimic environmental exposures that are thought to establish the immune cell milieu in humans, allowing for the detailed interrogation of molecular mechanisms of pulmonary macrophage adaptation. The importance is that the mechanisms that lead to lung-associated macrophages' functional and developmental adaptation remain elusive. Through a series of single-cell transcriptomics, high dimensional imaging, and flow cytometric characterization paired with in vivo and ex vivo challenge models, the authors identified a unique population of monocyte-derived alveolar macrophages that are Dectin-1 - Card9 signaling-dependent, that expressed high levels of CD11b, ApoE, were glycolytic, and produced large amounts of interleukin 6 upon restimulation. They showed that paracrine derived ApoE is critical for establishment of this population, dependent on M-CSF secretion. Functionally, the authors were able to demonstrate in vivo, that these β -glucan-elicited "MoAMs" limited the bacterial burden of Legionella pneumophila post-infection and ameliorated fibrosis severity in a murine

fibrosis model. Collectively these data identify MoAMs generated upon environmental cues and ApoE as an essential determinant for lung immune resilience. Together, their conclusions revealed that ApoE controls Dectin-1 -dependent development of monocyte-derived alveolar macrophages upon pulmonary b-glucan-induced inflammatory adaptation.

This paper is well written, well-argued, rigorous, novel, and poised to represent an important advance in the field.

Answer: We thank the reviewer for the positive, constructive feedback and work throughout the publication process.

Final Decision Letter:

Dear Dr. Schlitzer,

I am delighted to accept your manuscript entitled "Apolipoprotein E controls the development of monocyte-derived alveolar macrophages upon pulmonary inflammatory adaptation" for publication in an upcoming issue of Nature Immunology.

Over the next few weeks, your paper will be copyedited to ensure that it conforms to Nature Immunology style. Once your paper is typeset, you will receive an email with a link to choose the appropriate publishing options for your paper and our Author Services team will be in touch regarding any additional information that may be required.

Please note that *Nature Immunology* is a Transformative Journal (TJ). Authors may publish their research with us through the traditional subscription access route or make their paper immediately open access through payment of an article-processing charge (APC). Authors will not be required to make a final decision about access to their article until it has been accepted. Find out more about Transformative Journals.

Your paper will be published online soon after we receive your corrections and will appear in print in

the next available issue.

Also, if you have any spectacular or outstanding figures or graphics associated with your manuscript - though not necessarily included with your submission - we'd be delighted to consider them as candidates for our cover. Simply send an electronic version (accompanied by a hard copy) to us with a possible cover caption enclosed.

If you have not already done so, we strongly recommend that you upload the step-by-step protocols used in this manuscript to the Protocol Exchange. Protocol Exchange is an open online resource that allows researchers to share their detailed experimental know-how. All uploaded protocols are made freely available, assigned DOIs for ease of citation and fully searchable through nature.com. Protocols can be linked to any publications in which they are used and will be linked to from your article. You can also establish a dedicated page to collect all your lab Protocols. By uploading your Protocols to Protocol Exchange, you are enabling researchers to more readily reproduce or adapt the methodology you use, as well as increasing the visibility of your protocols and papers. Upload your Protocols at www.nature.com/protocolexchange/. Further information can be found at www.nature.com/protocolexchange/about .

Please note that we encourage the authors to self-archive their manuscript (the accepted version before copy editing) in their institutional repository, and in their funders' archives, six months after publication. Nature Portfolio recognizes the efforts of funding bodies to increase access of the research they fund, and strongly encourages authors to participate in such efforts. For information about our editorial policy, including license agreement and author copyright, please visit www.nature.com/ni/about/ed_policies/index.html

Sincerely,

Stephanie Houston, PhD
Senior Editor
Nature Immunology